# Unified View of Matrix Completion under General Structural Constraints

**Suriya Gunasekar**
UT at Austin, USA
suriya@utexas.edu

**Arindam Banerjee**
UMN Twin Cities, USA
banerjee@cs.umn.edu

**Joydeep Ghosh**
UT at Austin, USA
ghosh@ece.utexas.edu

## Abstract

Matrix completion problems have been widely studied under special low dimensional structures such as low rank or structure induced by decomposable norms. In this paper, we present a unified analysis of matrix completion under general low-dimensional structural constraints induced by *any* norm regularization. We consider two estimators for the general problem of structured matrix completion, and provide unified upper bounds on the sample complexity and the estimation error. Our analysis relies on generic chaining, and we establish two intermediate results of independent interest: (a) in characterizing the size or complexity of low dimensional subsets in high dimensional ambient space, a certain *partial* complexity measure encountered in the analysis of matrix completion problems is characterized in terms of a well understood complexity measure of Gaussian widths, and (b) it is shown that a form of restricted strong convexity holds for matrix completion problems under general norm regularization. Further, we provide several non-trivial examples of structures included in our framework, notably including the recently proposed spectral $k$-support norm.

## 1 Introduction

The task of completing the missing entries of a matrix from an incomplete subset of (potentially noisy) entries is encountered in many applications including recommendation systems, data imputation, covariance matrix estimation, and sensor localization among others. Traditionally ill–posed high dimensional estimation problems, where the number of parameters to be estimated is much higher than the number of observations, has been extensively studied in the recent literature. However, matrix completion problems are particularly ill–posed as the observations are both limited (high dimensional), and the measurements are extremely localized, i.e., the observations consist of individual matrix entries. The localized measurement model, in contrast to random Gaussian or sub–Gaussian measurements, poses additional complications in general high dimensional estimation.

For well–posed estimation in high dimensional problems including matrix completion, it is imperative that low dimensional structural constraints are imposed on the target. For matrix completion, the special case of low–rank constraint has been widely studied. Several existing work propose tractable estimators with near–optimal recovery guarantees for (approximate) low–rank matrix completion [8, 7, 28, 26, 18, 19, 22, 11, 20, 21]. A recent work [16] addresses the extension to structures with decomposable norm regularization. However, the scope of matrix completion extends for low dimensional structures far beyond simple low–rankness or decomposable norm structures.

In this paper, we consider a unified statistical analysis of matrix completion under a general set of low dimensional structures that are induced by *any* suitable norm regularization. We provide statistical analysis of two generalized matrix completion estimators, the *constrained norm minimizer*, and the *generalized matrix Dantzig selector* (Section 2.2). The main results in the paper (Theorem 1a–1b) provide unified upper bounds on the sample complexity and estimation error of these estimators

for matrix completion under *any* norm regularization. Existing results on matrix completion with low rank or other decomposable structures can be obtained as special cases of our general results.

Our unified analysis of sample complexity is motivated by recent work on high dimensional estimation using global (sub) Gaussian measurements [10, 1, 35, 3, 37, 5]. A key ingredient in the recovery analysis of high dimensional estimation involves establishing a certain variation of Restricted Isometry Property (RIP) [9] of the measurement operator. It has been shown that such properties are satisfied by Gaussian and sub–Gaussian measurement operators with high probability. Unfortunately, as has been noted before by Candes et al. [8], owing to highly localized measurements, such conditions are not satisfied in the matrix completion problem, and the existing results based on global (sub) Gaussian measurements are not directly applicable. In fact, a key question we consider is: given the radically limited measurement model in matrix completion, by how much would the sample complexity of estimation increase beyond the known sample complexity bounds for global (sub) Gaussian measurements. Our results upper bounds the sample complexity for matrix completion to within only a $\log d$ factor larger over that of global (sub) Gaussian measurements [10, 3, 5]. While the result is known for low rank matrix completion using nuclear norm minimization [26, 20], with a careful use of generic chaining, we show that the $\log d$ factor suffices for structures induced by *any* norm! As a key intermediate result, we show that a useful form *restricted strong convexity (RSC)* [27] holds for the localized measurements encountered in matrix completion under general norm regularized structures. The result substantially generalizes existing RSC results for matrix completion under the special cases of nuclear norm and decomposable norm regularization [26, 16].

For our analysis, we use tools from generic chaining [33] to characterize the main results (Theorem 1a–1b) in terms of the *Gaussian width* (Definition 1) of certain *error sets*. Gaussian widths provide a powerful geometric characterization for quantifying the complexity of a structured low dimensional subset in a high dimensional ambient space. Such a unified characterization in terms of Gaussian width has the advantage that numerous tools have been developed in the literature for bounding the Gaussian width for structured sets, and this literature can be readily leveraged to derive new recovery guarantees for matrix completion under suitable structural constraints (Appendix D.2).

In addition to the theoretical elegance of such a unified framework, identifying useful but potentially non–decomposable low dimensional structures is of significant practical interest. The broad class of structures enforced through symmetric convex bodies and symmetric atomic sets [10] can be analyzed under this paradigm (Section 2.1). Such specialized structures can potentially capture the constraints in certain applications better than simple low–rankness. In particular, we discuss in detail, a non–trivial example of the *spectral $k$–support norm* introduced by McDonald et al. [25].

To summarize the key contributions of the paper:
• Theorem 1a–1b provide unified upper bounds on sample complexity and estimation error for matrix completion estimators using general norm regularization: a substantial generalization of the existing results on matrix completion under structural constraints.
• Theorem 1a is applied to derive statistical results for the special case of matrix completion under spectral $k$–support norm regularization.
• An intermediate result, Theorem 5 shows that under any norm regularization, a form of Restricted Strong Convexity (RSC) holds in the matrix completion setting with extremely localized measurements. Further, a certain *partial* measure of complexity of a set is encountered in matrix completion analysis (12). Another intermediate result, Theorem 2 provides bounds on the *partial* complexity measures in terms of a better understood complexity measure of Gaussian width. These intermediate results are of independent interest beyond the scope of the paper.

**Notations and Preliminaries**

Indexes $i, j$ are typically used to index rows and columns respectively of matrices, and index $k$ is used to index the observations. $e_i$, $e_j$, $e_k$, etc. denote the standard basis in appropriate dimensions[1]. Notation $G$ ans $g$ are used to denote a matrix and vector respectively, with independent standard Gaussian random variables. $\mathbb{P}(.)$ and $\mathbb{E}(.)$ denote the probability of an event and the expectation of a random variable, respectively. Given an integer $N$, let $[N] = \{1, 2, \ldots, N\}$. Euclidean norm in a vector space is denoted as $\|x\|_2 = \sqrt{\langle x, x \rangle}$. For a matrix $X$ with singular values $\sigma_1 \geq \sigma_2 \geq \ldots$, common norms include the *Frobenius norm* $\|X\|_F = \sqrt{\sum_i \sigma_i^2}$, the *nuclear norm* $\|X\|_* = \sum_i \sigma_i$,

the *spectral norm* $\|X\|_{\mathrm{op}} = \sigma_1$, and the *maximum norm* $\|X\|_\infty = \max_{ij} |X_{ij}|$. Also let, $\mathbb{S}^{d_1 d_2 - 1} = \{X \in \mathbb{R}^{d_1 \times d_2} : \|X\|_F = 1\}$ and $\mathbb{B}^{d_1 d_2} = \{X \in \mathbb{R}^{d_1 \times d_2} : \|X\|_F \leq 1\}$. Finally, given a norm $\|.\|$ defined on a vectorspace $\mathcal{V}$, its *dual norm* is given by $\|X\|^* = \sup_{\|Y\| \leq 1} \langle X, Y \rangle$.

**Definition 1** (Gaussian Width). Gaussian width of a set $S \subset \mathbb{R}^{d_1 \times d_2}$ is a widely studied measure of complexity of a subset in high dimensional ambient space and is given by:
$$w_G(S) = \mathbb{E}_G \sup_{X \in S} \langle X, G \rangle, \tag{1}$$
where recall that $G$ is a matrix of independent standard Gaussian random variables. Some key results on Gaussian width are discussed in Appendix D.2.

**Definition 2** (Sub–Gaussian Random Variable [36]). The sub–Gaussian norm of a random variable $X$ is given by: $\|X\|_{\Psi_2} = \sup_{p \geq 1} p^{-1/2} (\mathbb{E}|X|^p)^{1/p}$. $X$ is said be $b$–*sub–Gaussian* if $\|X\|_{\Psi_2} \leq b$.

Equivalently, $X$ is sub–Gaussian if one of the following conditions are satisfied for some constants $k_1$, $k_2$, and $k_3$ [Lemma 5.5 of [36]].

(1) $\forall p \geq 1, (\mathbb{E}|X|^p)^{1/p} \leq b\sqrt{p}$,     (2) $\forall t > 0, \mathbb{P}(|X| > t) \leq e^{1 - t^2/k_1^2 b^2}$,

(3) $\mathbb{E}[e^{k_2 X^2/b^2}] \leq e$, or     (4) if $\mathbb{E}X = 0$, then $\forall s > 0, \mathbb{E}[e^{sX}] \leq e^{k_3 s^2 b^2/2}$.

**Definition 3** (Restricted Strong Convexity (RSC)). A function $\mathcal{L}$ is said to satisfy *Restricted Strong Convexity (RSC)* at $\Theta$ with respect to a subset $S$, if for some *RSC parameter* $\kappa_\mathcal{L} > 0$,
$$\forall \Delta \in S, \mathcal{L}(\Theta + \Delta) - \mathcal{L}(\Theta) - \langle \nabla \mathcal{L}(\Theta), \Delta \rangle \geq \kappa_\mathcal{L} \|\Delta\|_F^2. \tag{2}$$

**Definition 4** (Spikiness Ratio [26]). For $X \in \mathbb{R}^{d_1 \times d_2}$, a measure of the "spikiness" is given by:
$$\alpha_{\mathrm{sp}}(X) = \frac{\sqrt{d_1 d_2} \|X\|_\infty}{\|X\|_F}. \tag{3}$$

**Definition 5** (Norm Compatibility Constant [27]). The compatibility constant of a norm $\mathcal{R} : \mathcal{V} \to \mathbb{R}$ under a closed convex cone $\mathcal{C} \subset \mathcal{V}$ is defined as follows:
$$\Psi_\mathcal{R}(\mathcal{C}) = \sup_{X \in \mathcal{C} \setminus \{0\}} \frac{\mathcal{R}(X)}{\|X\|_F}. \tag{4}$$

## 2   Structured Matrix Completion

Denote the ground truth target matrix as $\Theta^* \in \mathbb{R}^{d_1 \times d_2}$; let $d = d_1 + d_2$. In the noisy matrix completion, observations consists of individual entries of $\Theta^*$ observed through an additive noise channel.

**Sub–Gaussian Noise:** Given, a list of independently sampled standard basis $\Omega = \{E_k = e_{i_k} e_{j_k}^\top : i_k \in [d_1], j_k \in [d_2]\}$ with potential duplicates, observations $(y_k) \in \mathbb{R}^{|\Omega|}$ are given by:
$$y_k = \langle \Theta^*, E_k \rangle + \xi \eta_k, \text{ for } k = 1, 2, \ldots, |\Omega|, \tag{5}$$
where $\eta \in \mathbb{R}^{|\Omega|}$ is the noise vector of independent sub–Gaussian random variables with $\mathbb{E}[\eta_k] = 0$, and $\|\eta_k\|_{\Psi_2} = 1$ (recall $\|.\|_{\Psi_2}$ from Definition 2), and $\xi^2$ is scaled variance of noise per observation, (note $\mathrm{Var}(\eta_k) \leq$ constant). Also, without loss of generality, assume normalization $\|\Theta^*\|_F = 1$.

**Uniform Sampling:** Assume that the entries in $\Omega$ are drawn independently and uniformly:
$$E_k \sim \mathrm{uniform}\{e_i e_j^\top : i \in [d_1], j \in [d_2]\}, \text{ for } E_k \in \Omega. \tag{6}$$

Given $\Omega$, define the linear operator $P_\Omega : \mathbb{R}^{d_1 \times d_2} \to \mathbb{R}^\Omega$ as follows ($e_k \in \mathbb{R}^{|\Omega|}$):
$$P_\Omega(X) = \sum_{k=1}^{|\Omega|} \langle X, E_k \rangle e_k \tag{7}$$

**Structural Constraints** For matrix completion with $|\Omega| < d_1 d_2$, low dimensional structural constraints on $\Theta^*$ are necessary for well–posedness. We consider a generalized constraint setting wherein for some low–dimensional *model space* $\mathcal{M}$, $\Theta^* \in \mathcal{M}$ is enforced through a surrogate *norm regularizer* $\mathcal{R}(.)$. We make no further assumptions on $\mathcal{R}$ other than it being a norm in $\mathbb{R}^{d_1 \times d_2}$.

**Low Spikiness** In matrix completion under uniform sampling model, further restrictions on $\Theta^*$ (beyond low dimensional structure) are required to ensure that the most informative entries of the matrix are observed with high probability [8]. Early work assumed stringent matrix incoherence conditions for low–rank completion to preclude such matrices [7, 18, 19], while more recent work [11, 26], relax these assumptions to a more intuitive restriction of the spikiness ratio, defined in (3). However, under this relaxation only an approximate recovery is typically guaranteed in low–noise regime, as opposed to near exact recovery under incoherence assumptions.

**Assumption 1** (Spikiness Ratio). *There exists $\alpha^* > 0$, such that*
$$\|\Theta^*\|_\infty = \alpha_{\mathrm{sp}}(\Theta^*) \frac{\|\Theta^*\|_F}{\sqrt{d_1 d_2}} \leq \frac{\alpha^*}{\sqrt{d_1 d_2}}. \qquad \square$$

## 2.1 Special Cases and Applications

We briefly introduce some interesting examples of structural constraints with practical applications.

**Example 1** (Low Rank and Decomposable Norms). *Low–rankness* is the most common structure used in many matrix estimation problems including collaborative filtering, PCA, spectral clustering, etc. Convex estimators using nuclear norm $\|\Theta\|_*$ regularization has been widely studied statistically [8, 7, 28, 26, 18, 19, 22, 11, 20, 21]. A recent work [16] extends the analysis of low rank matrix completion to general decomposable norms, i.e. $\mathcal{R} : \forall X, Y \in (\mathcal{M}, \mathcal{M}^\perp), \mathcal{R}(X+Y) = \mathcal{R}(X) + \mathcal{R}(Y)$.

**Example 2** (Spectral $k$–support Norm). A non–trivial and significant example of norm regularization that is not decomposable is the *spectral $k$–support* norm recently introduced by McDonald et al. [25]. Spectral $k$–support norm is essentially the vector $k$–support norm [2] applied on the singular values $\sigma(\Theta)$ of a matrix $\Theta \in \mathbb{R}^{d_1 \times d_2}$. Without loss of generality, let $\bar{d} = d_1 = d_2$.
Let $\mathcal{G}_k = \{g \subseteq [\bar{d}] : |g| \leq k\}$ be the set of all subsets $[\bar{d}]$ of cardinality at most $k$, and denote the set $\mathcal{V}(\mathcal{G}_k) = \{(v_g)_{g \in \mathcal{G}_k} : v_g \in \mathbb{R}^{\bar{d}}, \mathrm{supp}(v_g) \subseteq g\}$. The spectral $k$–support norm is given by:

$$\|\Theta\|_{\text{k–sp}} = \inf_{v \in \mathcal{V}(\mathcal{G}_k)} \Big\{ \sum_{g \in \mathcal{G}_k} \|v_g\|_2 : \sum_{g \in \mathcal{G}_k} v_g = \sigma(\Theta) \Big\}, \tag{8}$$

McDonald et al. [25] showed that spectral $k$–support norm is a special case of *cluster norm* [17]. It was further shown that in multi–task learning, wherein the tasks (columns of $\Theta^*$) are assumed to be clustered into dense groups, the cluster norm provides a trade–off between intra–cluster variance, (inverse) inter–cluster variance, and the norm of the task vectors. Both [17] and [25] demonstrate superior empirical performance of cluster norms (and $k$–support norm) over traditional trace norm and spectral elastic net minimization on bench marked matrix completion and multi–task learning datasets. However, there are no known work on the statistical analysis of matrix completion with spectral $k$–support norm regularization. In Section 3.2, we discuss the consequence of our main theorem for this non–trivial special case.

**Example 3** (Additive Decomposition). Elementwise sparsity is a common structure often assumed in high–dimensional estimation problems. However, in matrix completion, elementwise sparsity conflicts with Assumption 1 (and more traditional incoherence assumptions). Indeed, it is easy to see that with high probability most of the $|\Omega| \ll d_1 d_2$ uniformly sampled observations will be zero, and an informed prediction is infeasible. However, elementwise sparse structures are often used within an *additive decomposition* framework, wherein $\Theta^* = \sum_k \Theta^{(k)}$, such that each component matrix $\Theta^{(k)}$ is in turn structured (e.g. low rank+sparse used for robust PCA [6]). In such structures, there is no scope for recovering sparse components outside the observed indices, and it is assumed that: $\Theta^{(k)}$ is sparse $\Rightarrow \mathrm{supp}(\Theta^{(k)}) \subseteq \Omega$. Further, the sparsity assumption might still conflict with the spikiness assumption. In such cases, consistent matrix completion is feasible under additional regularity assumptions if the superposed matrix is non–spiky. A candidate norm regularizer for such structures is the weighted infimum convolution of individual structure inducing norms [6, 39],

$$\mathcal{R}_w(\Theta) = \inf \Big\{ \sum_k w_k \mathcal{R}_k(\Theta^{(k)}) : \sum_k \Theta^{(k)} = \Theta \Big\}.$$

**Example 4** (Other Applications). Other potential applications including *cut matrices* [30, 10], structures induced by *compact convex sets*, norms inducing *structured sparsity assumptions on the spectrum of* $\Theta^*$, etc. can also be handled under the paradigm of this paper.

## 2.2 Structured Matrix Estimator

Let $\mathcal{R}$ be the norm surrogate for the structural constraints on $\Theta^*$, and $\mathcal{R}^*$ denote its dual norm. We propose and analyze two convex estimators for the task of structured matrix completion:

**Constrained Norm Minimizer**

$$\widehat{\Theta}_{\text{cn}} = \operatorname*{argmin}_{\|\Theta\|_\infty \leq \frac{\alpha^*}{\sqrt{d_1 d_2}}} \mathcal{R}(\Theta) \qquad \text{s.t. } \|P_\Omega(\Theta) - y\|_2 \leq \lambda_{\text{cn}}. \tag{9}$$

**Generalized Matrix Dantzig Selector**

$$\widehat{\Theta}_{\text{ds}} = \operatorname*{argmin}_{\|\Theta\|_\infty \leq \frac{\alpha^*}{\sqrt{d_1 d_2}}} \mathcal{R}(\Theta) \qquad \text{s.t. } \frac{\sqrt{d_1 d_2}}{|\Omega|} \mathcal{R}^* P_\Omega^*(P_\Omega(\Theta) - y) \leq \lambda_{\text{ds}}, \tag{10}$$

where $P_\Omega^* : \mathbb{R}^\Omega \to \mathbb{R}^{d_1 \times d_2}$ is the linear adjoint of $P_\Omega$, i.e. $\langle P_\Omega(X), y \rangle = \langle X, P_\Omega^*(y) \rangle$.

**Note:** Theorem 1a–1b gives consistency results for (9) and (10), respectively, under certain conditions on the parameters $\lambda_{\mathrm{cn}} > 0$, $\lambda_{\mathrm{ds}} > 0$, and $\alpha^* > 1$. In particular, these conditions assume knowledge of the noise variance $\xi^2$ and spikiness ratio $\alpha_{\mathrm{sp}}(\Theta^*)$. In practice, both $\xi$ and $\alpha_{\mathrm{sp}}(\Theta^*)$ are typically unknown and the parameters are tuned by validating on held out data.

## 3   Main Results

We define the following "restricted" *error cone* and its subset:
$$\mathcal{T}_\mathcal{R} = \mathcal{T}_\mathcal{R}(\Theta^*) = \mathrm{cone}\{\Delta : \mathcal{R}(\Theta^* + \Delta) \leq \mathcal{R}(\Theta^*)\}, \text{ and } \mathcal{E}_\mathcal{R} = T_\mathcal{R} \cap \mathbb{S}^{d_1 d_2 - 1}. \tag{11}$$

Let $\widehat{\Theta}_{\mathrm{cn}}$ and $\widehat{\Theta}_{\mathrm{ds}}$ be the estimates from (9) and (10), respectively. If $\lambda_{\mathrm{cn}}$ and $\lambda_{\mathrm{ds}}$ are chosen such that $\Theta^*$ belongs to the feasible sets in (9) and (10), respectively, then the error matrices $\widehat{\Delta}_{\mathrm{cn}} = \widehat{\Theta}_{\mathrm{cn}} - \Theta^*$ and $\widehat{\Delta}_{\mathrm{ds}} = \widehat{\Theta}_{\mathrm{ds}} - \Theta^*$ are contained in $\mathcal{T}_\mathcal{R}$.

**Theorem 1a** (Constrained Norm Minimizer). *Under the problem setup in Section 2, let $\widehat{\Theta}_{cn} = \Theta^* + \widehat{\Delta}_{cn}$ be the estimate from (9) with $\lambda_{cn} = 2\xi$. For large enough $c_0$, if $|\Omega| > c_0^2 w_G^2(\mathcal{E}_\mathcal{R}) \log d$, then there exists an RSC parameter $\kappa_{c_0} > 0$ and constants $c_1$, $c_2$, $c_3$ such that with probability greater than $1 - \exp(-c_1 w_G^2(\mathcal{E}_\mathcal{R})) - 2\exp(-c_2 w_G^2(\mathcal{E}_\mathcal{R}) \log d)$,*

$$\frac{1}{d_1 d_2} \|\widehat{\Delta}_{cn}\|_F^2 \leq 4 \max \left\{ \frac{c_3 \xi^2}{\kappa_{c_0}}, \frac{\alpha^{*2}}{d_1 d_2} \frac{c_0^2 w_G^2(\mathcal{E}_\mathcal{R}) \log d}{|\Omega|} \right\}.$$

**Theorem 1b** (Matrix Dantzig Selector). *Under the problem setup in Section 2, let $\widehat{\Theta}_{ds} = \Theta^* + \widehat{\Delta}_{ds}$ be the estimate from (10) with $\lambda_{ds} \geq 2\xi \frac{\sqrt{d_1 d_2}}{|\Omega|} \mathcal{R}^* P_\Omega^*(w)$. For large enough $c_0$, if $|\Omega| > c_0^2 w_G^2(\mathcal{E}_\mathcal{R}) \log d$, then there exists an RSC parameter $\kappa_{c_0} > 1$, and constant $c_1$ such that with probability greater than $1 - \exp(-c_1 w_G^2(\mathcal{E}_\mathcal{R}))$,*

$$\|\widehat{\Delta}_{ds}\|_F^2 \leq 4 \max \left\{ \frac{\lambda_{ds}^2 \Psi_\mathcal{R}^2(\mathcal{T}_\mathcal{R})}{\kappa_{c_0}^2}, \alpha^{*2} \frac{c_0^2 w_G^2(\mathcal{E}_\mathcal{R}) \log d}{|\Omega|} \right\}.$$

Recall Gaussian width $w_G$ and subspace compatibility constant $\Psi_\mathcal{R}$ from (1) and (4), respectively.

**Remarks:**

1. If $\mathcal{R}(\Theta) = \|\Theta\|_*$ and $\mathrm{rank}(\Theta^*) = r$, then $w_G^2(\mathcal{E}_\mathcal{R}) \leq 3dr$, $\Psi_\mathcal{R}(\mathcal{T}_\mathcal{R}) \leq 2r$ and $\frac{\sqrt{d_1 d_2}}{|\Omega|} \|P_\Omega^*(\eta)\|_2 \leq 2\sqrt{\frac{d \log d}{|\Omega|}}$ w.h.p [10, 14, 26]. Using these bounds in Theorem 1b recovers near–optimal results for low rank matrix completion under spikiness assumption [26].

2. For both estimators, upper bound on sample complexity is dominated by the square of Gaussian width which is often considered the *effective dimension* of a subset in high dimensional space and plays a key role in high dimensional estimation under Gaussian measurement ensembles. The results show that, independent of $\mathcal{R}(.)$, the upper bound on sample complexity for consistent matrix completion with highly localized measurements is within a $\log d$ factor of the known sample complexity of $\sim w_G^2(\mathcal{E}_\mathcal{R})$ for estimation from Gaussian measurements [3, 10, 37, 5].

3. First term in estimation error bounds in Theorem 1a–1b scales with $\xi^2$ which is the per observation noise variance (upto constant). The second term is an upper bound on error that arises due to unidentifiability of $\Theta^*$ within a certain radius under the spikiness constraints [26]; in contrast [7] show exact recovery when $\xi = 0$ using more stringent matrix incoherence conditions.

4. Bound on $\widehat{\Delta}_{\mathrm{cn}}$ from Theorem 1a is comparable to the result by Candés et al. [7] for low rank matrix completion under non–low–noise regime, where the first term dominates, and those of [10, 35] for high dimensional estimation under Gaussian measurements. With a bound on $w_G^2(\mathcal{E}_\mathcal{R})$, it is easy to specialize this result for new structural constraints. However, this bound is potentially loose and asymptotically converges to a constant error proportional to the noise variance $\xi^2$.

5. The estimation error bound in Theorem 1b is typically sharper than that in Theorem 1a. However, for specific structures, using application of Theorem 1b requires additional bounds on $\mathbb{E} \mathcal{R}^*(P_\Omega(W))$ and $\Psi_\mathcal{R}(\mathcal{T}_\mathcal{R})$ besides $w_G^2(\mathcal{E}_\mathcal{R})$.

### 3.1   Partial Complexity Measures

Recall that for $w_G(S) = \mathbb{E} \sup_{X \in S} \langle X, G \rangle$ and $\mathbb{R}^{|\Omega|} \ni g \sim \mathcal{N}(0, I_{|\Omega|})$ is a standard normal vector.

**Definition 6** (Partial Complexity Measures)**.** *Given a randomly sampled collection* $\Omega = \{E_k \in \mathbb{R}^{d_1 \times d_2}\}$*, and a random vector* $\eta \in \mathbb{R}^{|\Omega|}$*, the partial* $\eta$*–complexity measure of S is given by:*

$$w_{\Omega,\eta}(S) = \mathbb{E}_{\Omega,\eta} \sup_{X \in S-S} \langle X, P_\Omega^*(\eta) \rangle. \tag{12}$$

The special cases where $\eta$ is a vector of standard Gaussian $g$, or standard Rademacher $\epsilon$ (i.e. $\epsilon_k \in \{-1,1\}$ w.p. $1/2$) variables, are of particular interest. In the case of symmetric $\eta$, like $g$ and $\epsilon$, $w_{\Omega,\eta}(S) = 2\mathbb{E}_{\Omega,\eta} \sup_{X \in S} \langle X, P_\Omega^*(\eta) \rangle$, and the later expression will be used interchangeably ignoring the constant term.

**Theorem 2** (Partial Gaussian Complexity)**.** *Let* $S \subset \mathbb{B}^{d_1 d_2}$*, and let* $\Omega$ *be sampled according to* (6)*.* $\exists$ *universal constants* $K_1$*,* $K_2$*,* $K_3$*, and* $K_4$ *such that:*

$$w_{\Omega,g}(S) \leq K_1 \sqrt{\frac{|\Omega|}{d_1 d_2}} w_G(S) + \min\left\{ K_2 \sqrt{\mathbb{E}_\Omega \sup_{X,Y \in S} \|P_\Omega(X-Y)\|_2^2}, K_3 \sup_{X \in S} \frac{\alpha_{sp}(X)}{\sqrt{d_1 d_2}} \right\} \tag{13}$$

*Further, for a centered i.i.d.* $1$*–sub–Gaussian vector* $\eta \in \mathbb{R}^{|\Omega|}$*,* $w_{\Omega,\eta}(S) \leq K_4 w_{\Omega,g}(S)$*.*

**Note:** For $\Omega \subsetneq [d_1] \times [d_2]$, the second term in (13) is a consequence of the localized measurements.

## 3.2 Spectral $k$–Support Norm

We introduced spectral $k$–support norm in Section 2.1. The estimators from (9) and (10) for spectral $k$–support norm can be efficiently solved through proximal methods using the proximal operators derived in [25]. We are interested in the statistical guarantees for matrix completion using spectral $k$–support norm regularization. We extend the analysis for upper bounding the Gaussian width of the descent cone for the vector $k$–support norm by [29] to the case of spectral $k$–support norm. WLOG let $d_1 = d_2 = \bar{d}$. Let $\sigma^* \in \mathbb{R}^{\bar{d}}$ be the vector of singular values of $\Theta^*$ sorted in non–ascending order. Let $r \in \{0,1,2,\dots,k-1\}$ be the unique integer satisfying: $\sigma^*_{k-r-1} > \frac{1}{r+1} \sum_{i=k-r}^p \sigma^*_i \geq \sigma^*_{k-r}$. Denote $I_2 = \{1,2,\dots,k-r-1\}$, $I_1 = \{k-r, k-r+1, \dots, s\}$, and $I_0 = \{s+1, s+2, \dots, \bar{d}\}$. Finally, for $I \subseteq [\bar{d}]$, $(\sigma^*_I)_i = 0 \;\forall i \in I^c$, and $(\sigma^*_I)_i = \sigma^*_i \;\forall i \in I$.

**Lemma 3.** *If rank of* $\Theta^*$ *is* $s$ *and* $\mathcal{E}_\mathcal{R}$ *is the error set from* $\mathcal{R}(\Theta) = \|\Theta\|_{k\text{-}sp}$*, then*

$$w_G^2(\mathcal{E}_\mathcal{R}) \leq s(2\bar{d}-s) + \left( \frac{(r+1)^2 \|\sigma^*_{I_2}\|_2^2}{\|\sigma^*_{I_1}\|_1^2} + |I_1| \right)(2\bar{d}-s).$$

Proof of the above lemma is provided in the appendix. Lemma 3 can be combined with Theorem 1a to obtain recovery guarantees for matrix completion under spectral $k$–support norm.

# 4 Discussions and Related Work

**Sample Complexity:** For consistent recovery in high dimensional convex estimation, it is desirable that the descent cone at the target parameter $\Theta^*$ is "small" relative to the feasible set (enforced by the observations) of the estimator. Thus, a measure of complexity/size of the error cone at $\Theta^*$ is crucial in establishing sample complexity and estimation error bounds. Results in this paper are largely characterized in terms of a widely used complexity measure of Gaussian width $w_G(.)$, and can be compared with the literature on estimation from Gaussian measurements.

**Error Bounds:** Theorem 1a provides estimation error bounds that depends only on the Gaussian width of the descent cone. In non–low–noise regime, this result is comparable to analogous results of constrained norm minimization [6, 10, 35]. However, this bound is potentially loose owing to mismatched data–fit term using squared loss, and asymptotically converges to a constant error proportional to the noise variance $\xi^2$.
A tighter analysis on the estimation error can be obtained for the matrix Dantzig selector (10) from Theorem 1b. However, application of Theorem 1b requires computing high probability upper bound on $\mathcal{R}^*(P_\Omega(W))$. The literature on norms of random matrices [13, 24, 36, 34] can be exploited in deriving such bounds. Beside, in special cases: if $\mathcal{R}(.) \geq K\|.\|_*$, then $K\mathcal{R}^*(.) \leq \|.\|_{op}$ can be used to obtain asymptotically consistent results.

Finally, under near zero–noise, the second term in the results of Theorem 1 dominates, and bounds are weaker than that of [6, 19] owing to the relaxation of stronger incoherence assumption.

**Related Work and Future Directions:** The closest related work is the result on consistency of matrix completion under decomposable norm regularization by [16]. Results in this paper are a strict generalization to general norm regularized (not necessarily decomposable) matrix completion. We provide non–trivial examples of application where structures enforced by such non–decomposable norms are of interest. Further, in contrast to our results that are based on Gaussian width, the RSC parameter in [16] depends on a modified complexity measure $\kappa_{\mathcal{R}}(d, |\Omega|)$ (see definition in [16]). An advantage of results based on Gaussian width is that, application of Theorem 1 for special cases can greatly benefit from the numerous tools in the literature for the computation of $w_G(.)$.

Another closely related line of work is the non–asymptotic analysis of high dimensional estimation under random Gaussian or sub–Gaussian measurements [10, 1, 35, 3, 37, 5]. However, the analysis from this literature rely on variants of RIP of the measurement ensemble [9], which is not satisfied by the the extremely localized measurements encountered in matrix completion[8]. In an intermediate result, we establish a form of RSC for matrix completion under general norm regularization: a result that was previously known only for nuclear norm and decomposable norm regularization.

In future work, it is of interest to derive matching lower bounds on estimation error for matrix completion under general low dimensional structures, along the lines of [22, 5] and explore special case applications of the results in the paper. We also plan to derive explicit characterization of $\lambda_{\mathrm{ds}}$ in terms of Gaussian width of unit balls by exploiting generic chaining results for general Banach spaces [33].

## 5 Proof Sketch

Proofs of the lemmas are provided in the Appendix.

### 5.1 Proof of Theorem 1

Define the following set of $\beta$–*non–spiky* matrices in $\mathbb{R}^{d_1 \times d_2}$ for constant $c_0$ from Theorem 1:

$$\mathbb{A}(\beta) = \left\{ X : \alpha_{\mathrm{sp}}(X) = \frac{\sqrt{d_1 d_2}\|X\|_\infty}{\|X\|_F} \leq \beta \right\}. \tag{14}$$

Define,
$$\beta_{c_0} = \frac{1}{c_0}\sqrt{\frac{|\Omega|}{w_G^2(\mathcal{E}_{\mathcal{R}})\log d}} \tag{15}$$

**Case 1: Spiky Error Matrix** When the error matrix from (9) or (10) has large spikiness ratio, following bound on error is immediate using $\|\widehat{\Delta}\|_\infty \leq \|\widehat{\Theta}\|_\infty + \|\Theta^*\|_\infty \leq 2\alpha^*/\sqrt{d_1 d_2}$ in (3).

**Proposition 4** (Spiky Error Matrix). *For the constant $c_0$ in Theorem 1a, if $\alpha_{sp}(\widehat{\Delta}_{cn}) \notin \mathbb{A}(\beta_{c_0})$, then* $\|\widehat{\Delta}_{cn}\|_F \leq 2c_0\alpha^*\sqrt{\frac{w_G^2(\mathcal{E}_{\mathcal{R}})\log d}{|\Omega|}}$. *An analogous result also holds for* $\widehat{\Delta}_{ds}$. □

**Case 2: Non–Spiky Error Matrix** Let $\widehat{\Delta}_{\mathrm{ds}}, \widehat{\Delta}_{\mathrm{cn}} \in \mathbb{A}(\beta)$.

#### 5.1.1 Restricted Strong Convexity (RSC)

Recall $\mathcal{T}_{\mathcal{R}}$ and $\mathcal{E}_{\mathcal{R}}$ from (11). The most significant step in the proof of Theorem 1 involves showing that over a useful subset of $\mathcal{T}_{\mathcal{R}}$, a form of RSC (2) is satisfied by a squared loss penalty.

**Theorem 5** (Restricted Strong Convexity). *Let $|\Omega| > c_0^2 w_G^2(\mathcal{E}_{\mathcal{R}})\log d$, for large enough constant $c_0$; further, sub–sampling excess samples such that $|\Omega| \sim \mathbf{\Omega}(w_G^2(\mathcal{E}_{\mathcal{R}})\log^2 d)$. There exists a RSC parameter $\kappa_{c_0} = 1 - \delta_{c_0} > 0$, such that the following holds w.p. greater that $1 - \exp(-c_1 w_G^2(\mathcal{E}_{\mathcal{R}}))$,*

$$\forall X \in \mathcal{T}_{\mathcal{R}} \cap \mathbb{A}(\beta), \quad \frac{d_1 d_2}{|\Omega|}\|P_\Omega(X)\|_2^2 \geq \kappa_{c_0}\|X\|_F^2.$$

Proof in Appendix A combines empirical process tools along with Theorem 2. □

Recall from (5), that $y - P_\Omega(\Theta^*) = \xi w$, where $w \in \mathbb{R}^{|\Omega|}$ consists of independent sub–Gaussian random variables with $\mathbb{E}[w_k] = 0$ and $\|w_k\|_{\Psi_2} = 1$ (recall $\|.\|_{\Psi_2}$ from Definition 2).

### 5.1.2 Constrained Norm Minimizer

**Lemma 6.** *Under the conditions of Theorem 1, let $c_1$ be a constant such that $\forall k$, $Var(w_k) \leq c_1$. $\exists$ a universal constant $c_2$ such that, if $\lambda_{cn} \geq 2c_1\xi\sqrt{|\Omega|}$, then with probability greater than $1 - 2\exp(-c_2|\Omega|)$, (a) $\widehat{\Delta}_{ds} \in \mathcal{T}_{\mathcal{R}}$, and (b) $\|P_\Omega(\widehat{\Delta}_{cn})\|_2 \leq 2\lambda_{cn}$.* □

Using $\lambda_{cn} = 2c_1\xi\sqrt{|\Omega|}$ in (9), if $\widehat{\Delta}_{cn} \in \mathbb{A}(\beta)$, then using Theorem 5 and Lemma 6, w.h.p.

$$\frac{\|\widehat{\Delta}_{cn}\|_F^2}{d_1 d_2} \leq \frac{1}{\kappa_{c_0}} \frac{\|P_\Omega(\widehat{\Delta}_{cn})\|_2^2}{|\Omega|} \leq \frac{4c_1^2\xi^2}{\kappa_{c_0}}. \tag{16}$$

### 5.1.3 Matrix Dantzig Selector

**Proposition 7.** $\lambda_{ds} \geq \xi \frac{\sqrt{d_1 d_2}}{|\Omega|} \mathcal{R}^* P_\Omega^*(w) \Rightarrow$ *w.h.p. (a) $\widehat{\Delta}_{ds} \in \mathcal{T}_{\mathcal{R}}$; (b) $\frac{\sqrt{d_1 d_2}}{|\Omega|} \mathcal{R}^* P_\Omega^*(P_\Omega(\widehat{\Delta}_{ds})) \leq 2\lambda_{ds}$.*

Above result follows from optimality of $\widehat{\Theta}_{ds}$ and triangle inequality. Also,

$$\frac{\sqrt{d_1 d_2}}{|\Omega|}\|P_\Omega(\widehat{\Delta}_{ds})\|_2^2 \leq \frac{\sqrt{d_1 d_2}}{|\Omega|} \mathcal{R}^* P_\Omega^*(P_\Omega(\widehat{\Delta}_{ds}))\mathcal{R}(\widehat{\Delta}_{ds}) \leq 2\lambda_{ds}\Psi_{\mathcal{R}}(\mathcal{T}_{\mathcal{R}})\|\widehat{\Delta}_{ds}\|_F,$$

where recall norm compatibility constant $\Psi_{\mathcal{R}}(\mathcal{T}_{\mathcal{R}})$ from (4). Finally, using Theorem 5, w.h.p.

$$\frac{\|\widehat{\Delta}_{ds}\|_F^2}{d_1 d_2} \leq \frac{1}{|\Omega|} \frac{\|P_\Omega(\widehat{\Delta}_{ds})\|_2^2}{\kappa_{c_0}} \leq \frac{4\lambda_{ds}\Psi_{\mathcal{R}}(\mathcal{T}_{\mathcal{R}})}{\kappa_{c_0}} \frac{\|\widehat{\Delta}_{ds}\|_F}{\sqrt{d_1 d_2}}. \tag{17}$$

## 5.2 Proof of Theorem 2

Let the entries of $\Omega = \{E_k = e_{i_k}e_{j_k}^\top : k = 1, 2, \ldots, |\Omega|\}$ be sampled as in (6). Recall that $g \in \mathbb{R}^{|\Omega|}$ is a standard normal vector. For a compact $S \subseteq \mathbb{R}^{d_1 \times d_2}$, it suffices to prove Theorem 2 for a dense countable subset of $S$. Overloading $S$ to such a countable subset, define following random process:

$$(\mathcal{X}_{\Omega,g}(X))_{X \in S}, \text{where } \mathcal{X}_{\Omega,g}(X) = \langle X, P_\Omega^*(g)\rangle = \sum_k \langle X, E_k\rangle g_k. \tag{18}$$

We start with a key lemma in the proof of Theorem 2. Proof of this lemma, provided in Appendix B, uses tools from the broad topic of generic chaining developed in recent works [31, 33].

**Lemma 8.** $\exists$ *constants $k_1$, $k_2$ such that for $S \subseteq \mathbb{S}^{d_1 d_2 - 1}$, then*

$$w_{\Omega,g}(S) = \mathbb{E} \sup_{X \in S} \mathcal{X}_{\Omega,g}(X) \leq k_1\sqrt{\frac{|\Omega|}{d_1 d_2}}w_G(S) + k_2\sqrt{\mathbb{E} \sup_{X,Y \in S}\|P_\Omega(X - Y)\|_2^2}. \qquad □$$

**Lemma 9.** *There exists constants $k_3$, $k_4$, such that for $S \subseteq \mathbb{S}^{d_1 d_2 - 1}$*

$$\mathbb{E} \sup_{X,Y \in S}\|P_\Omega(X - Y)\|_2^2 \leq k_3 \sup_{X \in S} \frac{\alpha_{sp}(X)}{\sqrt{d_1 d_2}}w_{\Omega,g}(S) + k_4\frac{|\Omega|}{d_1 d_2}w_G^2(S)$$

Theorem 2 follows by combining Lemma 8 and Lemma 9, and simplifying using $\sqrt{ab} \leq a/2 + b/2$ and triangle inequality (See Appendix B). The statement in Theorem 2 about partial sub–Gaussian complexity follows from a standard result in empirical process given in Lemma 12.

**Acknowledgments** We thank the anonymous reviewers for helpful comments and suggestions. S. Gunasekar and J. Ghosh acknowledge funding funding from NSF grants IIS-1421729, IIS-1417697, and IIS1116656. A. Banerjee acknowledges NSF grants IIS-1447566, IIS-1422557, CCF-1451986, CNS-1314560, IIS-0953274, IIS-1029711, and NASA grant NNX12AQ39A.

## Footnotes

[1]for brevity we omit the explicit dependence of dimension unless necessary

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
