[Supplementary Material]

# Supplementary Material: Unified View of Matrix Completion under General Structural Constraints

Suriya Gunasekar, Arindam Banerjee, Joydeep Ghosh

Note: Background and preliminaries are provided in Appendix D.

## A Appendix to Proof of Theorem 1

### A.1 Proof of Theorem 5

**Statement of Theorem 5**:
Let $|\Omega| > c_0^2 w_G^2(\mathcal{E}_\mathcal{R}) \log d$, for large enough constant $c_0$. There exists a RSC parameter $\kappa_{c_0} > 0$ with $\kappa_{c_0} \approx 1 - o\left(\frac{1}{\sqrt{\log d}}\right)$, and a constant $c_1$ such that, the following holds w.p. greater that $1 - \exp(-c_1 w_G^2(\mathcal{E}_\mathcal{R}))$,

$$\forall X \in \mathcal{T}_\mathcal{R} \cap \mathbb{A}(\beta_{c_0}), \quad \frac{d_1 d_2}{|\Omega|} \|P_\Omega(X)\|_2^2 \geq \kappa_{c_0} \|X\|_F^2.$$

**Proof:** Recall that $\mathcal{T}_\mathcal{R} = \{\Delta : \mathcal{R}(\Theta^* + \Delta) \leq \mathcal{R}(\Theta^*)$ and $\mathcal{E}_\mathcal{R} = \mathcal{T}_\mathcal{R} \cap \mathbb{S}^{d_1 d_2 - 1}$. Using the properties of norms, it can be easily verified that for the non–trivial case of $\Theta^* \neq 0$, $\mathcal{T}_\mathcal{R}$ is a cone with non–empty interior.

We use Theorem 2 as a key result in this proof.
Define $\bar{\mathcal{E}}_\mathcal{R} = \mathcal{T}_\mathcal{R} \cap \mathbb{B}^{d_1 d_2}$.
$\bar{\mathcal{E}}_\mathcal{R}$ is a compact subset of $\mathcal{T}_\mathcal{R}$ with non–empty interior, which satisfies the conditions of Theorem 2. Also, since $\mathcal{T}_\mathcal{R} \cap \mathbb{A}(\beta_{c_0})$ is a cone, the following can be easily verified:

$$\begin{aligned} w_{\Omega,g}(\bar{\mathcal{E}}_\mathcal{R} \cap \mathbb{A}(\beta_{c_0})) &= w_{\Omega,g}(\mathcal{E}_\mathcal{R} \cap \mathbb{A}(\beta_{c_0})) \\ w_G(\bar{\mathcal{E}}_\mathcal{R} \cap \mathbb{A}(\beta_{c_0})) &= w_G(\mathcal{E}_\mathcal{R} \cap \mathbb{A}(\beta_{c_0})) \leq w_G(\mathcal{E}_\mathcal{R}) \end{aligned} \tag{19}$$

We define a random variable $V(\Omega) = \sup_{X \in \mathcal{E}_\mathcal{R} \cap \mathbb{A}(\beta_{c_0})} \left| \frac{d_1 d_2}{|\Omega|} \|P_\Omega(X)\|_2^2 - 1 \right|$.

Note that: $\mathbb{E} \frac{d_1 d_2}{|\Omega|} \|P_\Omega(X)\|^2 = 1$; and
for $X \in \mathbb{A}(\beta_{c_0})$, $\|X\|_\infty \leq \frac{\beta_{c_0}}{\sqrt{d_2 d_2}}$.

#### A.1.1 Expectation of $V(\Omega)$

Recall that $\Omega = \{E_k : s = 1, 2, \dots |\Omega|\}$ are sampled uniformly form standard basis for $\mathbb{R}^{d_1 \times d_2}$, $(\epsilon_k)$ are a sequence of independent Rademacher variables, and $w_G(.)$ denotes the Gaussian width. For constant $k_1, k_2, k_3$ not necessarily same in each occurrence:

$$\begin{aligned} \mathbb{E} V(\Omega) &\overset{(a)}{\leq} \frac{2 d_1 d_2}{|\Omega|} \mathbb{E} \sup_{X \in \mathcal{E}_\mathcal{R} \cap \mathbb{A}(\beta_{c_0})} \left| \sum_{k=1}^{|\Omega|} \langle X, E_k \rangle^2 \epsilon_k \right| \overset{(b)}{\leq} k_1 \beta_{c_0} \frac{\sqrt{d_1 d_2}}{|\Omega|} \mathbb{E} \sup_{X \in \mathcal{E}_\mathcal{R} \cap \mathbb{A}(\beta_{c_0})} \left| \sum_{k=1}^{|\Omega|} \langle X, E_k \rangle \epsilon_k \right| \\ &= k_1 \beta_{c_0} \frac{\sqrt{d_1 d_2}}{|\Omega|} w_{\Omega,\epsilon}(\bar{\mathcal{E}}_\mathcal{R} \cap \mathbb{A}(\beta_{c_0})) \overset{(c)}{\leq} k_1 \sqrt{\frac{\beta_{c_0}^2 w_G^2(\mathcal{E}_\mathcal{R})}{|\Omega|}} + k_2 \frac{\beta_{c_0}^2}{|\Omega|} \leq \frac{k_3}{c_0 \sqrt{\log d}}, \quad (20) \end{aligned}$$

where $(a)$ follows from symmetrization (Lemma 18), $(b)$ from contraction principle as $\phi_k(\langle X, E_k \rangle) = \frac{\langle X, E_k \rangle^2}{2 \sup_{X \in \mathcal{E}_\mathcal{R} \cap \mathbb{A}(\beta_{c_0})} \|X\|_\infty}$ is a contraction (Lemma 19), and $(c)$ follows from Theorem 2.

### A.1.2 Concentration about $\mathbb{E}V(\Omega)$

Let $\Omega' \subset [m] \times [n]$ be another set of indices that differ from $\Omega$ in exactly one element. We have:

$$V(\Omega) - V(\Omega') = \sup_{X \in \mathcal{E}_\mathcal{R} \cap \mathbb{A}(\beta_{c_0})} \left| \frac{d_1 d_2}{|\Omega|} \sum_{ij \in \Omega} X_{ij}^2 - 1 \right| - \sup_{X \in \mathcal{E}_\mathcal{R} \cap \mathbb{A}(\beta_{c_0})} \left| \frac{d_1 d_2}{|\Omega|} \sum_{kl \in \Omega'} X_{kl}^2 - 1 \right|$$

$$\leq \frac{d_1 d_2}{|\Omega|} \sup_{X \in \mathcal{E}_\mathcal{R} \cap \mathbb{A}(\beta_{c_0})} \left( \left| \sum_{ij \in \Omega} X_{ij}^2 - \sum_{kl \in \Omega'} X_{kl}^2 \right| \right)$$

$$\leq \frac{2 d_1 d_2}{|\Omega|} \sup_{X \in \mathcal{E}_\mathcal{R} \cap \mathbb{A}(\beta_{c_0})} \|X\|_\infty^2 \leq \frac{2\beta_{c_0}^2}{|\Omega|}. \tag{21}$$

By similar arguments on $V(\Omega') - V(\Omega)$, $|V(\Omega) - V(\Omega')| \leq \frac{2\beta_{c_0}^2}{|\Omega|}$. Therefore, using Mc Diarmid's inequality (34), we have $P(V(\Omega) > \mathbb{E}V(\Omega) + \delta) \leq \exp\left(-c_1' \frac{\delta^2 |\Omega|}{\beta_{c_0}^4}\right)$. Using $\delta = \frac{1}{c_0 \sqrt{\log d}}$, we have

$$P\left( V(\Omega) > \frac{k_3'}{c_0 \sqrt{\log d}} \right) \leq \exp\left( - c_1 w_G^2(\mathcal{E}_\mathcal{R}) \right),$$

where $c_0$ is a constant that can be chosen independent of $k_3$. Choosing $c_0$ large enough, we can set $\kappa_{c_0} := 1 - \delta_{c_0} = 1 - \frac{k_3'}{c_0 \sqrt{\log d}}$ close to 1. $\qquad\square$

### A.2 Proof of Lemma 6

Recall that $\eta \in \mathbb{R}^{|\Omega|}$ is a vector of centered, unit variance sub-Gaussian random variables with $\|\eta_k\|_{\Psi_2} \leq b$. Combining Lemma 25 and Lemma 26, we have that $\eta_k^2$ and $\eta_k^2 - 1$ are sub–exponential with $\|\eta_k^2 - 1\|_{\Psi_1} \leq 2\|\eta_k^2\|_{\Psi_1} \leq 4\|\eta_k\|_{\Psi_2} \leq 4b^2$. Thus, using Lemma 24, for a constant $c_2'$, we have:

$$\mathbb{P}\left( \left| \frac{1}{|\Omega|} \sum_{k=1}^{|\Omega|} \eta_k^2 - 1 \right| > \tau \right) \leq 2\exp\left( - c_2' |\Omega| \min\left\{ \frac{\tau^2}{16b^4}, \frac{\tau}{4b^2} \right\} \right). \tag{22}$$

Choosing $\tau$ to be an appropriate constant, we have $\|P_\Omega(\Theta^*) - y\|_2 \leq 2\xi\sqrt{|\Omega|} \leq \lambda_{\text{cn}}$ w.p. greater than $1 - \exp(-c_2\tau|\Omega|)$, and the lemma follows from the optimality of $\widehat{\Theta}_{\text{cn}}$ and triangle inequality.

## B Appendix to Proof of Theorem 2

### B.1 Results from Generic Chaining

In this section, $K$ denotes a universal constant, not necessarily the same at each occurrence.

**Definition 7** (Gamma Functional (Definition 2.2.19 in [33])). Given a complete pseudometric space $(T, d)$, an *admissible sequence* is an increasing sequence $(\mathcal{A}_n)$ of partitions of $T$ such that $|\mathcal{A}_0| = 1$ and $|\mathcal{A}_n| \leq 2^{2^n}$ for $n \geq 1$. For $\alpha > 0$, we define the Gamma functional $\gamma_\alpha(T, d)$ as follows:

$$\gamma_\alpha(T, d) = \inf_{(\mathcal{A}_n)_{n \geq 0}} \sup_{t \in T} \sum_{n \geq 0} 2^{n/\alpha} \Delta_d(A_n(t)), \tag{23}$$

where inf is over all admissible sequences $(\mathcal{A}_n)$, $A_n(t)$ is the unique element of $\mathcal{A}_n$ that contains $t$, and $\Delta_d(A)$ is the diameter of the set $A$ measured in metric $d$.

**Lemma 10** (Majorizing Measures Theorem (Theorem 2.4.1 in [33])). *Given a closed set $T$ in a metric space, let $(X_t)_{t \in T}$ be a centered Gaussian process indexed by $t \in T$, i.e. $(X_t)$ are jointly Gaussian. For $s, t \in T$, let $d_X(s, t) := \sqrt{\mathbb{E}(X_s - X_t)^2}$ denote the canonical pseudometric associated with $(X_t)$. We then have :*

$$\frac{1}{K} \gamma_2(T, d_X) \leq \mathbb{E}\sup_{t \in T} X_t \leq L\gamma_2(T, d_X).$$

*In particular, considering the canonical Gaussian process $(\sum_i t_i g_i)_{t \in T}$, we have:*

$$\frac{1}{K}\gamma_2(T, \|.\|_F) \le w_G(T) \le K\gamma_2(T, \|.\|_F).$$

**Lemma 11** (Theorem 2.4.12 in [33]). *Let $(X_t)_{t \in T}$ be a centered Gaussian process with canonical distance $d_X = \sqrt{\mathbb{E}(X_s - X_t)^2}$. Let $(Y_t)_{t \in T}$ be another centered process indexed by the same set $T$, such that it satisfies the following condition:*

$$\forall s, t \in T, u > 0, \quad \mathbb{P}(|Y_s - Y_t| > u) \le 2\exp\Big(-\frac{u^2}{2d_X^2(s,t)}\Big),$$

*then, we have $\mathbb{E}\sup_{s,t \in T}|Y_s - Y_t| \le K\mathbb{E}\sup_{t \in T}X_t$.*

*If further, $(Y_t)_{t \in T}$ is symmetric, then $\mathbb{E}\sup_t |Y_t| \le \mathbb{E}\sup_{s,t \in T}|Y_s - Y_t| = 2\mathbb{E}\sup_{t \in T}Y_t$.*

*Note that from the properties of sub–Gaussian random variables, the above lemma can be directly bound canonical sub–Gaussian complexity measures using canonical Gaussian complexities.*

**Lemma 12** (Theorem 3.1.4 in [33]). *Let $T$ be a compact group with non–empty interior. Consider a translation invariant random distance $d_\omega$, that depends on a random parameter $\omega$ and let $d(s,t) = \sqrt{\mathbb{E}d_\omega^2(s,t)}$, then :*

$$\big(\mathbb{E}\gamma_2^2(T, d_\omega)\big)^{1/2} \le K\gamma_2(T, d) + K\big(\mathbb{E}\sup_{s,t \in T}d_\omega^2(s,t)\big)^{1/2}$$

## B.2 Proof of Lemma 8

**Statement of Lemma 8**
For a compact subset $S \subseteq \mathbb{R}^{d_1 \times d_2}$ with non–empty interior, $\exists$ constants $k_1$, $k_2$ such that:

$$w_{\Omega,g}(S) = \mathbb{E}\sup_{X \in S}\mathcal{X}_{\Omega,g}(X) \le k_1\sqrt{\frac{|\Omega|}{d_1 d_2}}w_G(S) + k_2\sqrt{\mathbb{E}\sup_{X,Y \in S}\|P_\Omega(X-Y)\|_2^2}. \qquad \square$$

**Proof:** Recall definition of $(\mathcal{X}_{\Omega,g}(X))_{X \in S}$ from (18), such that $\mathcal{X}_{\Omega,g}(X) = \sum_k \langle X, E_k\rangle g_k$.

By Fubini's theorem $\mathbb{E}_{\Omega,g}\sup_{X \in S}\mathcal{X}_{\Omega,g}(X) = \mathbb{E}_\Omega\mathbb{E}_g\sup_{X \in S}\mathcal{X}_{\Omega,g}(X)$.

Also, we have the following results:

- For a fixed $\Omega$, $(\mathcal{X}_{\Omega,g}(X))$ is a Gaussian process with a translation invariant canonical distance given by $d_\Omega(X,Y) = \|P_\Omega(X-Y)\|_2^2$.
- $d(X,Y) := \sqrt{E_\Omega d_\Omega^2(X,Y)} = \sqrt{\frac{|\Omega|}{d_1 d_2}}\|X-Y\|_F$

Using Lemma 10 we have: $\mathbb{E}_g\sup_{X \in S}\mathcal{X}_{\Omega,g}(X) \le K\gamma_2(S, d_\Omega)$, and the following holds:

$$w_{\Omega,g}(S) = \mathbb{E}_\Omega\mathbb{E}_g\sup_{X \in S}\mathcal{X}_{\Omega,g}(X) \le K\mathbb{E}_\Omega\gamma_2(S, d_\Omega) \overset{(a)}{\le} \sqrt{\mathbb{E}_\Omega\gamma_2^2(S, d_\Omega)}$$

$$\overset{(b)}{\le} K\sqrt{\frac{|\Omega|}{d_1 d_2}}\gamma_2(S, \|.\|_F) + K\sqrt{\mathbb{E}\sup_{X,Y \in S}\|P_\Omega(X-Y)\|_2^2}, \qquad (24)$$

where $(a)$ follows from Jensen's inequality, $(b)$ from Lemma 12 and noting that by definition $\forall M > 0$, $\gamma_2(T, M\dot{d}) = M\gamma_2(T, d)$. Lemma 8 now follows from (24) and Lemma 10. $\qquad \square$

## B.3 Proof of Lemma 9

**Statement of Lemma 9**
There exists constants $k_3$, $k_4$, such that for compact $S \subseteq \mathbb{B}^{d_1 d_2}$ with non–empty interior

$$\mathbb{E}\sup_{X,Y \in S}\|P_\Omega(X-Y)\|_2^2 \le k_3\frac{|\Omega|}{d_1 d_2}w_G^2(S) + k_4\sup_{X,Y \in S}\|X-Y\|_\infty w_{\Omega,g}(S)$$

**Proof:** Using triangle inequality, we have:

$$\mathbb{E}\sup_{X,Y\in S}\|P_\Omega(X-Y)\|_2^2 \leq \mathbb{E}\sup_{X,Y\in S}\big|\|P_\Omega(X-Y)\|_2^2 - \mathbb{E}\|P_\Omega(X-Y)\|_2^2\big| + \sup_{X,Y\in S}\mathbb{E}\|P_\Omega(X-Y)\|_2^2 \tag{25}$$

Further,

$$\sup_{X,Y\in S}\mathbb{E}\|P_\Omega(X-Y)\|_2^2 = \frac{|\Omega|}{d_1 d_2}\sup_{X,Y}\|X-Y\|_F^2 \leq \frac{|\Omega|}{d_1 d_2}\gamma_2^2(S,\|.\|_F)^2, \tag{26}$$

where the last inequality follows from the definition of $\gamma_\alpha$.

Finally, we have the following set of equations:

$$\mathbb{E}\sup_{X,Y\in S}\big|\|P_\Omega(X-Y)\|_2^2 - \mathbb{E}[\|P_\Omega(X-Y)\|_2^2]\big| = \mathbb{E}\sup_{X,Y\in S}\big|\sum_{k=1}^{|\Omega|}\langle X-Y,E_k\rangle^2 - \mathbb{E}\langle X-Y,E_k\rangle^2\big|$$

$$\overset{(a)}{\leq} 2\mathbb{E}_{\Omega,(\epsilon_s)}\sup_{X,Y\in S}\big|\sum_{k=1}^{|\Omega|}\langle X-Y,E_k\rangle^2\epsilon_k\big| \overset{(b)}{\leq} k_4' \sup_{X\in S}\|X-Y\|_\infty \mathbb{E}_{\Omega,g}\sup_{X,Y\in S}\big|\sum_{k=1}^{|\Omega|}\langle X-Y,E_k\rangle g_k\big|$$

$$\overset{(c)}{\leq} 2k_4' \sup_{X,Y\in S}\|X-Y\|_\infty \mathbb{E}_{\Omega,g}\sup_{X\in S}\big|\sum_{k=1}^{|\Omega|}\langle X,E_k\rangle g_k\big| \overset{(d)}{\leq} 4k_4'\sup_{X,Y\in S}\|X-Y\|_\infty w_{\Omega,g}(S), \tag{27}$$

where $(\epsilon_k)$ are standard Rademacher variables, i.e. $\epsilon_k\in\{-1,1\}$ with equal probability, $(a)$ follows from symmetrization argument (Lemma 18), $(b)$ follows from contraction principles Lemma 19 and using $\phi(\langle X,E_k\rangle) = \frac{\langle X,E_k\rangle^2}{2\sup_{X\in S}\|X\|_\infty}$ as a contraction, $(c)$ follows from triangle inequality, and $(d)$ follows from $g_k$ being symmetric (Lemma 2.2.1 in [33]). $\qquad\square$

The lemma follows by combining Lemma 10 and equations (25), (26), and (27).

## B.4 Remaining Steps in the Proof of Theorem 2

From Lemma9, we have the following:

$$\sqrt{\mathbb{E}\sup_{X,Y\in S}\|P_\Omega(X-Y)\|_2^2} \overset{(a)}{\leq} K_3\sqrt{\frac{|\Omega|}{d_1 d_2}}w_G(S) + \sqrt{k_4(\sup_{X,Y\in S}\|X-Y\|_\infty)w_{\Omega,g}(S)}$$

$$\overset{(b)}{\leq} K_3\sqrt{\frac{|\Omega|}{d_1 d_2}}w_G(S) + K_4(\sup_{X,Y\in S}\|X-Y\|_\infty) + \frac{1}{2}w_{\Omega,g}(S), \tag{28}$$

where $(a)$ follows from triangle inequality, $(b)$ using $\sqrt{ab}\leq a/2 + b/2$.

Bound on $w_{\Omega,g}(S)$ in Theorem 2 follows by using (28) in Lemma 8.

## C Spectral k–Support Norm

Recall the following definition of spectral $k$–support norm $\|\Theta\|_{\text{k–sp}}$ from (8):

$$\|\Theta\|_{\text{k–sp}} = \inf_{v\in\mathcal{V}(\mathcal{G}_k)}\Big\{\sum_{g\in\mathcal{G}_k}\|v_g\|_2 : \sum_{g\in\mathcal{G}_k}v_g = \sigma(\Theta)\Big\}, \tag{29}$$

where $\mathcal{G}_k = \{g\subseteq[\bar{d}] : |g|\leq k\}$ is the set of all subsets $[\bar{d}]$ of cardinality at most $k$, and $\mathcal{V}(\mathcal{G}_k) = \{(v_g)_{g\in\mathcal{G}_k} : v_g\in\mathbb{R}^{d_1}, \text{supp}(v_g)\subseteq g\}$.

**Proposition 13** (Proposition 2.1 in [2]). *For* $\Theta\in\mathbb{R}^{\bar{d}\times\bar{d}}$ *with singular values* $\sigma(\Theta) = \{\sigma_1,\sigma_2,\ldots,\sigma_{\bar{d}}\}$, *such that* $\sigma_1\geq\sigma_2\geq\ldots,\geq\sigma_{\bar{d}}$. *Then,*

$$\|\Theta\|_{k-sp} = \left(\sum_{i=1}^{k-r-1}\sigma_i^2 + \frac{1}{r+1}\left(\sum_{i=k-r}^{\bar{d}}\sigma_i\right)^2\right)^{\frac{1}{2}}, \tag{30}$$

*where* $r\in\{0,1,2,\ldots,k-1\}$ *is the unique integer satisfying* $\sigma_{k-r-1} > \frac{1}{r+1}\sum_{i=k-r}^{d_1}\sigma_i \geq \sigma_{k-r}.\square$

## C.1 Proof of Lemma 3

**Statement of Lemma 3**
If rank of $\Theta^*$ is $s$ and $\mathcal{E}_\mathcal{R}$ is the error set from $\mathcal{R}(\Theta) = \|\Theta\|_{\text{k-sp}}$, then

$$w_G^2(\mathcal{E}_\mathcal{R}) \leq s(2\bar{d} - s) + \left( \frac{(r+1)^2 \|\sigma_{I_2}^*\|_2^2}{\|\sigma_{I_1}^*\|_1^2} + |I_1| \right)(2\bar{d} - s).$$

$\square$

**Proof** We state the following lemmas from existing work.

**Lemma 14** (Equation **60** in [29]). *Let $z$ be an $s \geq k$ sparse vector in $\mathbb{R}^p$, and let $\tilde{z}$ is the vector $z$ sorted in non increasing order of $|z_i|$. Denote $r \in \{0, 1, 2, \ldots, k-1\}$ to be the unique integer satisfying*

$$|\tilde{z}_{k-r-1}| > \frac{1}{r+1} \sum_{i=k-r}^p |\tilde{z}_i| \geq |\tilde{z}_{k-r}|.$$

*Define $I_2 = \{1, 2, \ldots, k-r-1\}$, $I_1 = \{k-r, k-r+1, \ldots, s\}$, and $I_0 = \{s+1, s+2, \ldots, p\}$; and let $\tilde{z}_I$ denote the vector $\tilde{z}$ restricted to indices in $I$. Then the sub–differential of the vector $k$–support norm denoted by $\|.\|_{\text{vk-sp}}$ at $w$ is given by:*

$$\partial \|z\|_{\text{vk-sp}} = \frac{1}{\|z\|_{\text{vk-sp}}} \left\{ \tilde{z}_{I_2} + \frac{1}{r+1} \|\tilde{z}_{I_1}\|_1 (sign(\tilde{z}_{I_1}) + h_{I_0}) : \|h\|_\infty \leq 1 \right\},$$

**Lemma 15** (Theorem 2 in [38]). *Let $\mathcal{R} : \mathbb{R}^{d_1 \times d_2} \to \mathbb{R}_+$ be an orthogonally invariant norm; i.e. $\mathcal{R}(X) = \phi(\sigma(X))$ such that $\phi : \mathbb{R}^{d_1} \to \mathbb{R}_+$ is a symmetric gauge function satisfying: (a) $\phi(x) > 0 \ \forall x \neq 0$, (b) $\phi(\alpha x) = |\alpha| \phi(x)$, (c) $\phi(x + y) \leq \phi(x) + \phi(y)$, and (d) $\phi(x) = \phi(|x|)$.*

*Further let $\partial \phi(x)$ denote the sub–differential of $\phi$ at $x$. Then for $X \in \mathbb{R}^{\bar{d} \times \bar{d}}$ with singular value decomposition (SVD) $X = U_X \Sigma_X V_X^\top$ and $\sigma_X = diag(\Sigma_X)$, the sub–differential of $\mathcal{R}(X)$ is given by:*

$$\partial \mathcal{R}(X) = \{U_X D V_X^\top : D = diag(d), \text{ and } d \in \partial \Phi(\sigma_X)\}.$$

Since spectral $k$–support norm of a matrix $X = U_X \Sigma_X V_X^\top$ is the vector $k$–support norm applied to the singular values $\sigma_X = \text{diag}(\Sigma_X)$, Lemma 14 and 15 can be used to infer the following:

$$\partial \|X\|_{\text{k-sp}} = \left\{ U_X D V_X^\top : \text{diag}(D) \in \frac{1}{\|\sigma_X\|_{\text{vk-sp}}} \left\{ \sigma_{X_{I_2}} + \frac{\|\sigma_{X_{I_1}}\|_1}{r+1} (\mathbf{1}_{I_1} + h_{I_0}) : \|h\|_\infty \leq 1 \right\} \right\}.$$

(31)

where $\mathbf{1} \in \mathbb{R}^{\bar{d}}$ denotes a vector of all ones.

The error cone for $\mathcal{R}(.) = \|.\|_{\text{k-sp}}$ is given by the tangent cone:

$$\mathcal{T}_\mathcal{R} = \text{cone}\{\Delta : \|\Theta^* + \Delta\|_{\text{k-sp}} \leq \|\Theta^*\|_{\text{k-sp}}\},$$

and the polar of the tangent cone – the *normal cone* is given by

$$\mathcal{T}_\mathcal{R}^* = \mathcal{N}_\mathcal{R}(\Theta^*) = \{Y : \langle Y, X \rangle \leq 0 \ \forall X \in \mathcal{T}_\mathcal{R}\} = \text{cone}(\partial \mathcal{R}(\Theta^*))$$

Let $\Theta^* = U^* \Sigma^* V^{*\top}$ be the full SVD of $\Theta^*$, such that $\sigma^* = \text{diag}(\Sigma^*) \in \mathbb{R}^{\bar{d}}$ and $\sigma_1^* \geq \sigma_2^* \ldots \geq \sigma_{\bar{d}}^*$. Let $u_i^*$ and $v_i^*$ for $i \in [\bar{d}]$ denote the $i^{\text{th}}$ column of $U^*$ and $V^*$, respectively. Further, let the rank of $\Theta^*$ be $\text{rk}(\Theta^*) = \|\sigma^*\|_0 = s$.

Like for the vector case, denote $r \in \{0, 1, 2, \ldots, k-1\}$ to be the unique integer satisfying $\sigma_{k-r-1}^* > \frac{1}{r+1} \sum_{i=k-r}^p \sigma_i^* \geq \sigma_{k-r}^*$. Define $I_2 = \{1, 2, \ldots, k-r-1\}$, $I_1 = \{k-r, k-r+1, \ldots, s\}$, and $I_0 = \{s+1, s+2, \ldots, p\}$; Also define the subspace:

$$T = \text{span}\{u_i^* x^\top : i \in I_2 \cup I_1, x \in \mathbb{R}^{\bar{d}}\} \cup \text{span}\{y v_i^{*\top} : i \in I_2 \cup I_1, y \in \mathbb{R}^{\bar{d}}\}$$

Let $T^\perp$ be the subspace orthogonal to $T$ and let $P_T$ and $P_{T^\perp}$ be the projection operators onto $T$ and $T^\perp$, respectively. From (31) we have,

$$\mathcal{N}_{\mathcal{R}}(\Theta^*) = \left\{ Y = U^* D V^{*\top} : D = \text{diag}\left( t \frac{r+1}{\|\sigma_{I_1}^*\|_1} \sigma_{I_2}^* + t\mathbf{1}_{I_1} + th_{I_0} \right) : t \geq 0, \|h\|_\infty \leq 1 \right\},$$

Finally, from Lemma 21, we have that

$$w_G^2(\mathcal{T}_{\mathcal{R}} \cap \mathbb{S}^{\bar d \bar d - 1}) \leq \mathbb{E}_G \inf_{X \in \mathcal{N}_{\mathcal{R}}(\Theta^*)} \|G - X\|_F^2$$

$$\leq \mathbb{E}_G \inf_{\substack{t > 0 \\ \|h\|_\infty \leq 1}} \left\| P_T(G) - t \frac{r+1}{\|\sigma_{I_1}^*\|_1} \sum_{i \in I_2} \sigma_i^* u_i^* v_i^{*\top} + t \sum_{i \in I_1} u_i^* v_i^{*\top} + P_{T^\perp}(G) - t \sum_{i \in I_0} h_i u_i^* v_i^{*\top} \right\|_F^2$$

Let $P_{T^\perp}(G) = \sum_{i \in I_0} \sigma_i(P_{T^\perp}G) u_i^* v_i^{*\top}$ be the decomposition of $P_T^\perp(G)$ in the basis of of $\{u_i^* v_i^{*\top}\}_{i \in I_0}$. Taking $t = \|P_{T^\perp}(G)\|_{\text{op}} = \max_{i \in I_0} \sigma_i(P_{T^\perp}(G))$, and $h_i = \sigma_i(P_{T^\perp}(G)) / \|P_{T^\perp}(G)\|_{\text{op}} \leq 1$, we have:

$$w_G^2(\mathcal{T}_{\mathcal{R}} \cap \mathbb{S}^{\bar d \bar d - 1}) \leq \mathbb{E}_G \|P_T(G)\|_F^2 + \left( \frac{(r+1)^2 \|\sigma_{I_2}^*\|_2^2}{\|\sigma_{I_1}^*\|_1^2} + |I_1| \right) \mathbb{E}_G \|P_T(G)\|_2^2. \tag{32}$$

Lemma 3 follows by using $\mathbb{E}_G \|P_T(G)\|_F^2 = s(2\bar d - s)$ and $\mathbb{E}_G \|P_T(G)\|_{\text{op}}^2 \leq 2(2\bar d - s)$ from [10].

# D Preliminaries

## D.1 Probability and Concentration

**Lemma 16** (Bernstein's Inequality (moment version)). *Let $X_i, i = 1, 2, \ldots, N$ be independent zero mean random variables. Further, let $\sigma^2 = \sum_i \mathbb{E}[X_i^2]$, and $M > 0$ be such that the following moment conditions are satisfied for $p \geq 2$,*

$$\mathbb{E}[X_i^p] \leq \frac{p! \sigma^2 M^{p-2}}{2}$$

*Then the following concentration inequality holds:*

$$\mathbb{P}\left( \left| \sum_i X_i \right| > u \right) \leq 2 \exp\left( \frac{-u^2}{2\sigma^2 + 2Mu} \right) \tag{33}$$

**Lemma 17** (McDiarmid's Inequality). *Let $X_i, i = 1, 2, \ldots, N$ be independent random variables. Consider a function $f : \mathbb{R}^N \to \mathbb{R}$:*

$$\text{If} \quad \forall i, \quad \sup_{X_1, X_2, \ldots, X_N, X_i'} |f(X_1, X_2, \ldots, X_N) - f(X_1, X_2, \ldots, X_{i-1}, X_i', X_{i+1}, \ldots, X_N)| \leq c_i,$$

$$\text{then,} \quad \mathbb{P}(|f(X_1, X_2, \ldots, X_N) - \mathbb{E}f(X_1, X_2, \ldots, X_N)| > u) \leq 2 \exp\left( \frac{-2u^2}{\sum_i c_i^2} \right) \tag{34}$$

**Lemma 18** (Symmetrization (Lemma 6.3 in [23])). *Let $F : \mathbb{R}_+ \to \mathbb{R}_+$ be a convex function, and $X_i, i = 1, 2, \ldots$ be a sequence of mean zero random variables in a Banach space $B$, s.t $\forall i, \mathbb{E}F\|X_i\| < \infty$. Denote a vector of standard Rademacher variables of appropriate dimension as $(\epsilon_i)$, then*

$$\mathbb{E}F\left( \frac{1}{2} \| \sum_i \epsilon_i X_i \| \right) \leq \mathbb{E}F\| \sum_i X_i \| \leq \mathbb{E}F\left( 2\| \sum_i \epsilon_i X_i \| \right) \tag{35}$$

*Further, if $X_i$ are not centered, then $\mathbb{E}F\left( \| \sum_i X_i - \mathbb{E}[X_i]\| \right) \leq \mathbb{E}F\left( 2\| \sum_i \epsilon_i X_i \| \right)$*

**Lemma 19** (Contraction Principle). *Consider a bounded $T \subset \mathbb{R}^N$, a standard Gaussian and standard Rademacher sequence, $(g_i) \in \mathbb{R}^N$ and $(\epsilon_i) \in \mathbb{R}^N$, respectively. If $\phi_i : \mathbb{R} \to \mathbb{R}, i \leq N$ are contractions, i.e. $\forall s, t \in \mathbb{R}, |\phi_i(s) - \phi_i(t)| \leq |s - t|$, and with $\phi_i(0) = 0$, then for any convex*

*function $F : \mathbb{R}_+ \to \mathbb{R}_+$, the following results are from Corollary 3.17, Theorem 4.12, and Lemma 4.5, respectively in [23]:*

$$\mathbb{E}F\Big(\frac{1}{2}\sup_{t\in T}\Big|\sum_{i=1}^{N}g_i\phi_i(t_i)\Big|\Big) \leq \mathbb{E}F\Big(2\sup_{t\in T}\Big|\sum_{i=1}^{N}g_it_i\Big|\Big) \tag{36}$$

$$\mathbb{E}F\Big(\frac{1}{2}\sup_{t\in T}\Big|\sum_{i=1}^{N}\epsilon_i\phi_i(t_i)\Big|\Big) \leq \mathbb{E}F\Big(2\sup_{t\in T}\Big|\sum_{i=1}^{N}\epsilon_it_i\Big|\Big) \tag{37}$$

$$\mathbb{E}F\Big(\|\sum_{i=1}^{N}\epsilon_it_i\|\Big) \leq \mathbb{E}F\Big(\sqrt{\frac{\pi}{2}}\|\sum_{i=1}^{N}g_it_i\|\Big) \tag{38}$$

## D.2 Gaussian Width

Gaussian width plays a key role high dimensional estimation, and plenty of tools have been developed for computing Gaussian widths of compact subsets [12, 23, 33, 10]. The existing work is specially well adapted for computing Gaussian widths for intersection of convex cones with unit norm balls [10], and recent work of [3] propose a mechanism for exploiting these tools for arbitrary compact sets. We briefly note some of the key results that aid in computing Gaussian widths. Recall that $\mathbb{S}^{d_1d_2-1}$ is a unit Euclidean sphere in $\mathbb{R}^{d_1\times d_2}$. Further, for a cone $\mathcal{C} \in \mathbb{R}^{d_1\times d_2}$, we define the polar cone as $\mathcal{C}^\circ = \{X : \langle X, Y\rangle \leq 0, \forall Y \in \mathcal{C}\}$.

### D.2.1 Direct Estimation

The Gaussian width of a compact set $T$ can be directly estimated as a supremum of Gaussian process over dense countable subset $\bar{T}$ of $T$ as $w_G(T) = \sup_{X\in\bar{T}}\langle X, G\rangle$.

We state the following properties are often used in direct estimation. These properties are consolidated from [33], [10] and [3]. In the following statements, $k$ is a constant not necessarily the same in each occurrence:

- Translation invariant and homogeneous: for any $a \in \mathbb{R}$, $w_G(S + a) = w_G(S)$; and .

- $w_G(\text{conv}(T)) \leq w_G(T)$

- $w_G(T_1 + T_2) \leq w_G(T_1) + w_G(T_2)$

- If $T_1 \subseteq T_2$, then $w_G(T_1) \leq w_G(T_2)$.

- If $T_1$ and $T_2$ are convex, then $w_G(T_1 \cup T_2) + w_G(T_1 \cap T_2) = w_G(T_1) + w_G(T_2)$

### D.2.2 Dudley's Inequality and Sudakov Minorization

**Definition 8** (Covering Number). Consider a metric $d$ defined on $S \subset \mathbb{R}^{d_1\times d_2}$. Given $\epsilon > 0$, the $\epsilon$–covering number of $S$ with respect to $d$, denoted by $\mathcal{N}(S, \epsilon, d)$, is the minimum number of points $\{\bar{X}_1, \bar{X}_2, \ldots, \bar{X}_{\mathcal{N}(S,\epsilon,d)}\}$ such that $\forall\ X \in S$, there exists $i \in \{1, 2, \ldots, \mathcal{N}(S, \epsilon, d)\}$ with $d(X, \bar{X}_i) \leq \epsilon$. The set $\{\bar{X}_1, \bar{X}_2, \ldots, \bar{X}_{\mathcal{N}(S,\epsilon,d)}\}$ is called the $\epsilon$–cover of $S$.

**Lemma 20** (Dudley's Inequality and Sudakov Minoration). *If $S$ is compact, then for any $\epsilon > 0$, there exists a constant $c$ s. t.*

$$c\epsilon\sqrt{\log \mathcal{N}(S, \epsilon, \|.\|_F)} \leq w_G(S) \leq 24\int_0^\infty \sqrt{\mathcal{N}(S, \epsilon, \|.\|_F)}d\epsilon.$$

*The upper bound is the Dudley's inequality and lower bound is by Sudakov minoration.*

### D.2.3 Geometry of Polar Cone

**Lemma 21** (Proposition **3.6** and Theorem **3.9** of [10]). *If $\mathcal{C} \subset \mathbb{R}^{d_1\times d_2}$ is a non–empty convex cone and $\mathcal{C}^\circ$ be its polar cone, then:*

Distance to polar cone : $w_G(\mathcal{C} \cap \mathbb{S}^{d_1d_2-1}) \leq \mathbb{E}_G[\inf_{X\in\mathcal{C}^\circ}\|G - X\|_F]$

Volume of polar cone : $w_G(\mathcal{C} \cap \mathbb{S}^{d_1 d_2 - 1}) \leq 3\sqrt{\dfrac{4}{\mathrm{vol}(C^\circ \cap \mathbb{S}^{d_1 d_2 - 1})}}$

### D.2.4 Infimum over Translated Cones

**Lemma 22** (Lemma **3** of [3]). *Let $S \subset \mathbb{R}^{d_1 \times d_2}$, and given $X \in S$, define $\rho(X) = \sup_{Y \in S} \|X - Y\|_F$ as the diameter of $S$ measured along $X$. Also define $\mathcal{G}(X) = cone(S - X) \cap \rho(X)\mathbb{B}^{d_1 d_2}$, where $\mathbb{B}^{d_1 d_2}$ is the unit Euclidean ball. Then,*
$$w_G(S) \leq \inf_{X \in S} w_G(\mathcal{G}(X))$$

### D.2.5 Generic Chaining

Lemma 10 (from [33]) gives the tightest bounds on the Gaussian width of a set. The definition of $\gamma_2$ (23) can be used derive tight bounds on the Gaussian width that are optimal upto constants. Further results and examples on using $\gamma$–functionals for Gaussian width computation can be found in the works of Talagrand [31, 32, 33].

## D.3 Sub–Gaussian and Sub–Exponential Random Variables

Recall the definition of sub–Gaussian random variables from Definition 2.

**Definition 9** (Sub–Exponential Random Variables). A random variable $X$ is said be *sub-exponential* if it satisfies one of the following equivalent conditions for $k_1$, $k_2$, and $k_3$ differing from one other by constants [Definition 5.13 of [36]].

1. $\mathbb{P}(|X| > t) \leq e^{1 - t/k_1}, \forall\, t > 0$,

2. $\forall p \geq 1,\ (\mathbb{E}[|X|^p])^{1/p} \leq k_2 p$, or

3. $\mathbb{E}[e^{X/k_3}] \leq e$.

The *sub–exponential norm* is given by:

$$\|X\|_{\Psi_1} = \inf\left\{ t > 0 : \mathbb{E}\exp\left(\frac{|X|}{t}\right) \leq 2 \right\} = \sup_{p \geq 1} p^{-1}(\mathbb{E}[|X|^p])^{1/p}. \tag{39}$$

The following results on sub–Gaussian and sub–exponential variables are from [36].

**Lemma 23** (Hoeffding–type inequality, Proposition **5.10** in [36]). *Let $X_1, X_2, \ldots, X_N$ be independent centered sub-Gaussian random variables, and let $K = \max_i \|X_i\|_{\Psi_2}$. Then, $\forall a \in \mathbb{R}^N$ and $t \geq 0$, $\exists$ constant $c$ s.t.,*

$$\mathbb{P}\left( \Big| \sum_{i=1}^N a_i X_i \Big| \geq t \right) \leq 2\exp\left(\frac{-ct^2}{K^2 \|a\|_2^2}\right). \tag{40}$$

**Lemma 24** (Bernstein–type inequality, Proposition **5.16** in [36]). *Let $X_1, X_2, \ldots, X_N$ be independent centered sub-exponential random variables, and let $K = \max_i \|X_i\|_{\Psi_1}$. Then $\forall a \in \mathbb{R}^N$, and $t \geq 0$, there exists a constant $c$ s.t.*

$$\mathbb{P}\left( \Big| \sum_{i=1}^N a_i X_i \Big| \geq t \right) \leq 2\exp\left( -c\min\left\{ \frac{t^2}{K^2 \|a\|_2^2}, \frac{t}{K \|a\|_\infty} \right\} \right). \tag{41}$$

**Lemma 25** (Lemma **5.14** in [36]). *$X$ is sub–Gaussian if and only if $X^2$ is sub–exponential. Further, $\|X\|_{\Psi_2}^2 \leq \|X^2\|_{\Psi_1} \leq 2\|X\|_{\Psi_2}^2$.*

**Lemma 26** (Remark **5.18** in [36]). *If $X$ is sub–Gaussian (or sub–exponential), then so is $X - \mathbb{E}X$. Further, $\|X - \mathbb{E}X\|_{\Psi_2} \leq 2\|X\|_{\Psi_2}$; $\quad \|X - \mathbb{E}X\|_{\Psi_1} \leq 2\|X\|_{\Psi_1}$.*

# E  Extension to GLMs

This section provides directions for extending the work to matrix completion under generalized linear models. This section has not been rigorously formalized. An accurate version will be included in a longer version of the paper.

We consider an observation model wherein the observation matrix $Y$ is drawn from a member of *natural exponential family* parametrized by a structured ground truth matrix $\Theta^*$, such that:

$$P(Y|\Theta^*) = \prod_{ij} p(Y_{ij}) \, e^{Y_{ij}\Theta_{ij}^* - A(\Theta_{ij}^*)}, \qquad (42)$$

where $A : \text{dom}(\Theta_{ij}) \to \mathbb{R}$ is called the *log–partition function* and is strictly convex and analytic, and $p(.)$ is called the *base measure*. This family of distributions encompass a wide range of common distributions including Gaussian, Bernoulli, binomial, Poisson, and exponential among others. In a generalized linear matrix completion setting [16], the task is to estimate $\Theta^*$ from a subset of entries $\Omega$ of $Y$, i.e. $(\Omega, P_\Omega(Y))$.

A useful consequence of exponential family distribution assumption for observation matrix is that the negative log–likelihood loss over the observed entries is convex with respect to the natural parameter $\Theta^*$, and have a one-to-one correspondence with a rich class of divergence functions called the Bregman Divergence [15, 4]. The negative log likelihood is proportional to:

$$\mathcal{L}_\Omega(\Theta) = \sum_{(i,j)\in\Omega} A(\Theta_{ij}) - Y_{ij}\Theta_{ij}$$

We propose the following *regularized matrix estimator* for generalized matrix completion:

$$\widehat{\Theta}_{\text{re}} = \operatorname*{argmin}_{\|\Theta\|_\infty \le \frac{\alpha^*}{\sqrt{d_1 d_2}}} \frac{d_1 d_2}{|\Omega|} \mathcal{L}_\Omega(\Theta) + \lambda_{\text{re}} \mathcal{R}(\Theta). \qquad (43)$$

**Hypothesis 1.** *Let $\widehat{\Theta}_{re} = \Theta^* + \widehat{\Delta}_{re}$. In addition to the assumptions in Section 2, we assume that for some $\eta \ge 0$, $\nabla^2 A(u) \ge e^{-\eta|u|} \forall\, u \in \mathbb{R}$. The following result holds for any fixed $\gamma > 1$. We define:*

$$\widetilde{\mathcal{T}}_{\mathcal{R},\gamma} = cone\{\Delta : \mathcal{R}(\Theta^* + \Delta) \le \mathcal{R}(\Theta^*) + \frac{1}{\gamma}\mathcal{R}(\Theta^*)\}, \quad and \quad \widetilde{\mathcal{E}}_{\mathcal{R},\gamma} = \widetilde{\mathcal{T}}_{\mathcal{R},\gamma} \cap \mathbb{S}^{d_1 d_2 - 1}. \quad (44)$$

*Let $\lambda_{re} \ge \gamma \frac{d_1 d_2}{|\Omega|} \mathcal{R}^*(\nabla \mathcal{L}_\Omega(\Theta^*))$, and for some $c_0$, $|\Omega| > \left(\frac{\gamma+1}{\gamma-1}\right)^2 c_0^2 w_G^2(\widetilde{\mathcal{E}}_{\mathcal{R},\gamma}) \log d$. There exists a constant $k_1$ such that for large enough $c_0$, there exists $\kappa_{c_0} > 0$, such that with high probability,*

$$\|\widehat{\Delta}_{re}\|_F^2 \le 4\alpha^{*2}\left(\frac{\gamma+1}{\gamma-1}\right)^2 \max\left\{\frac{\lambda_{re}^2 \Psi_{\mathcal{R}}^2(\widetilde{\mathcal{T}}_{\mathcal{R},\gamma})}{\zeta(\eta,\alpha^*)\kappa_{c_0}^2}, \frac{c_0^2 w_G^2(\widetilde{\mathcal{E}}_{\mathcal{R},\gamma}) \log d}{|\Omega|}\right\},$$

*where $\zeta(\eta,\alpha^*) = e^{\frac{-4\eta\alpha^*}{\sqrt{d_1 d_2}}}$, and $\alpha^*$, $w_G(.)$, and $\Psi_{\mathcal{R}}(.)$ are notations from Section 3.*

The conjectures follows by combining the results in this paper along with the results from [3], and [16]. This result is beyond the scope of this paper and will be dealt with more rigorously in a longer version of the paper.