[Reviews · NeurIPS 2015]

Submitted by Assigned_Reviewer_1

The paper studies matrix estimation problems from noisy subsampled entries (matrix completion) with general regularization and gives statistical guarantees that depend on the gaussian width of the tangent cone of the true parameter, thus formally connecting the subsampling measurement model to the random gaussian measurement model. This also generalizes the work on matrix completion type problems.

The authors propose two different estimators, one related to the dantzig selector and one more closely related to the constrained form of the lasso. The Dantzig selector estimator has better statistical performance (better dependence on noise level) but also requires knowledge of more structural parameters.

The main technical ingredients are a connection between a complexity term arising more naturally from the subsampling model and the gaussian width and verification of an RSC condition under more general regularizers. The connection between complexity measures is interesting and may have applications elsewhere. I am curious about the tightness of Theorem 2. Is there a way to lower bound the new complexity measure by the gaussian width?

I do have several high-level questions: 1. The dependence on |\Omega| in the success probabilities in Theorems 1a-1b seem strange to me, especially considering that my error rate gets clipped at the noise level. If |\Omega| is large then the success probability is very small while the error does not decrease. Is there a good explanation for this? 2. I am also curious about the dependence on the noise level in Theorem 1a. Since the observations are sampled i.i.d. with replacement, consistent recovery should be possible as |\Omega| \rightarrow \infty, but Theorem 1a does not demonstrate this phenomenon. Just as a sanity check, existing results (see Negahban and Wainwright for example) do demonstrate consistency with the number of samples. Is there a reason why it is not possible here?

3. Actually Theorem 1b also does not guarantee consistent recovery for fixed dimension as |\Omega| \rightarrow \infty since the success probability drops to zero with |\Omega|. Consistency is easy here as you can just average all of our observations but I am wondering why the theorems here do not default to this setting?
Summary: The paper gives statistical guarantees on noisy matrix completion-type problems with any norm regularizer in terms of the gaussian width of an appropriate error cone. This significantly generalizes existing results and I think enables many new applications to high dimensional estimation with subsampled observations but I am unsure about how best to interpret the main theorems (see questions below).

Submitted by Assigned_Reviewer_2

This paper introduces a general analysis of matrix completion with norm regularization. The analysis setting can be thought of as a generalization of the setting of Negahban and Wainwright, in which we assume that the target matrix has small L-\infty norm, and attempt to prove that natural optimization formulations stably recover the target matrix, up to an error which is proportional to the maximum of the noise level and a floor (which is a consequence of only assuming the target is not spiky).

The paper motivates this problem through several examples, including spectral-k-support norms, and norms for cut matrices. It states results showing that the estimation error and number of samples required is controlled by the squared Gaussian width of the descent cone of the regularizer at the target, times a logarithmic oversampling factor. This has a nice interpretation - namely, that "in general" the number of measurements in the standard basis only needs to be a logarithmic factor larger than the number of Gaussian measurements.

While it is not clear how mainstream some of the examples are, it is certainly welcome to have general tools which show that accurate recovery from elementwise subsamples is not a phenomenon which is restricted to the nuclear norm.

Smaller issues

The main theorem becomes vacuous when the number of sampled entries approaches w_G(E_R)^4. This seems to mostly be a technical issue, but it should be corrected. For general norms w_G could be quite small, and taking w_G^4 samples could represent an interesting scenario.

There are imprecisions in the mathematics. The calculations in the supplement seem to move back and forth between a Bernoulli sampling model (which is implicitly used in A.1.1, (18), (a) - otherwise || PO[ X ] ||_F^2 is not a sum of independent random variables, due to collisions) and the uniform sampling model, which is used elsewhere.

Proposition 4: \alpha_sp( \hat{\Delta} ) \in A(c_0, E_R ). Should just be \hat{\Delta} \in A(c_0, E_R).

While the paper analyzes matrix completion in any norm (and hence with any prior information that can be encoded in a norm), there are some norms that are not compatible with the spikiness condition. For example, the paper gives robust PCA as an example of an additive decomposition that could be analyzed using the tools here. There exist analyses of robust PCA with missing entries, but all that can be shown is that observed portion of the sparse component is recovered.

Unless I'm mistaken, it should be possible to prove a better bound than (12) on the quantity w_{\Omega,W}(S), using a fairly simple argument. For S of interest here, w_G(S) > 1, and so the bound (12) is larger than w_G(S) (often substantially larger). One can write (abbreviating \Omega as O and P_\Omega as PO):

w_{O,W}(S) = E_O

E_W sup_{X \in S-S} < X, PO(W) >

(11)

<= 2 E_O E_W sup_{X \in S} < X, PO(W) >

=

2 E_O w_G( PO(S) ),

(*)

Consider the natural metric space on S, with d(s,t) = ( E | < s - t, w > |^2 )^(1/2); projection onto \Omega is a contraction, with respect to this distance and so it does not increase the \gamma_2 function, hence,

w_G(PO(S)) <= K w_G(S),

giving

w_{O,W}(S) <= K' w_G(S) via (*) and the fact that \gamma_2 gives a tight bound on the expected supremum of a Gaussian process. Am I missing something?

The authors should be careful about anonymizing their supplementary material for conference submissions.
Summary: The paper proves results on the estimation of "non-spiky" matrices from random subsets of their entries, using norm regularization. The results give stable estimation in noise, and pertain to "any" norm, allowing unconventional regularizers. While the motivating examples considered here may not be so mainstream, the analysis is quite general.

Submitted by Assigned_Reviewer_3

This paper looks at general signal recovery using matrix completion kind of measurements. That is, when entries of the signal are sampled at random. It shows that under a spikiness condition on the signal general signals can be recovered with near minimal number of measurements.

Some comments:

- "Early work assumed stringent matrix incoherence conditions to preclude such matrices [8, 19, 20], while more recent work [11, 27], relax these assumptions to a more intuitive restriction of the spikiness ratio"

This is not an accurate statement since with spikiness assumption one can not get exact recovery when the noise go to zero. Where as with the incoherence of [8,19,20] one can. I agree that spikiness is less restrictive but the results are significantly weaker. In particular note that one can show (e.g. see Chatterjee) that just a simple SVD can match the results of nuclear norm when using the spikiness assumption. This would incorrectly suggest that that a simple SVD is enough for solving the matrix completion. In practice it is not. To me the test of a good result for matrix completion is:

1) if the noise is zero do you get exact recovery? 2) in the presence of noise is the rate good?

I would note that (1) is actually rather important because it is well understood that for such problems the low noise regime is the really difficult case.

Lines 84-87 Again as I mentioned these claims are not accurate. It only generalizes a particular class of results.

Summary: This paper looks at general signal recovery using matrix completion kind of measurements. That is, when entries of the signal are sampled at random. Recovering general signal structure with this form of measurements is interesting. My only gripe with this paper is that it does not look at traditional incoherence conditions which in my view make the final results significantly weaker.

Submitted by Assigned_Reviewer_4

The paper proposes a framework for

matrix completion under general structural constraints, which are encoded by minimizing corresponding matrix norms.

The main result of the authors is proving a bound on the number of samples required to recover a matrix with low norm from randomly drawn entries in terms of the Gaussian width - which is only a logarithmic factor away from the optimal number of measurements needed using Gaussian sensing.

This shows that in general, under 'non-spikiness' conditions on the unknown matrix X, one does not lose too much information by observing only single entries.

The authors also specialize their results for the spectral k-support norm for which known efficient proximal methods exist.

The result seems novel and interesting. The paper is entirely theoretical, and it is hard to asses its practical importance -

there are no simulations/real-data analysis, and the theoretical guarantees include constants which may be large in practice.

The authors formulate two convex optimization problems, but it's not clear how easy it is to solve these in practice - there is no algorithm proposed except for mentioning a solver for the specialized k-support norm.

Minor: the authors mention decomposable norm but don't define it anywhere

Summary: Very nice theoretical generalization of recovery results for matrix completion under general structural constraints. Not clear how tight are the bounds achieved and their relevant to practical recovery performance

Submitted by Assigned_Reviewer_5

This paper presents a unified view of matrix completion with structural constraints as regularization, thereby generalizing over a number of existing techniques such as low-rank completion, spectral k-support norm, and additive decomposition.

Summary: I feel that I do not appreciate the significance of the results. The unification itself seems quite trivial, I have seen similar formulations before, although not expressed as directly. The theoretical properties seem generalizations of existing results, and the importance of the contributions were not clear.

Partly the paper layout is to blame. The paper is quite heavy with the details of the approach in the first two pages, before the task definition and preliminaries are introduced, with minimal discussion at the end (except for Section 4). I had to go back and forth a few times to the introduction section to better appreciate the results.

Submitted by Assigned_Reviewer_6

The paper considers the problem of matrix completion, why uniform support and sub-gaussian noise. The original matrix is assumed to have low norm value with respect to a general regulariser.

The paper provides, for two estimators based on norm minimisation using L2 and dantzig-like constraints, respectively, L2 error bounds on the estimation error. The bounds are expressed using the Gaussian width of a cone associated to the norm.

The results are similar to the body of recent works on recovery of arbitrary structures from sub-Gaussian linear measurements, although in the case here of masked noisy measurements.

The error bounds have an extra log \d factor compared to the case of linear measurements. This is hard to tell from the paper whether this is inherent to the completion setting, or due to the analysis, and this question is left open.

Clarity -------- I find the presentation a bit dry. In particular:

- The starting point of the paper seem to be that usual results for linear inverse problems with gaussian measurements do not apply for matrix completion. This is mentioned without proof, reference or informal explanation.

- In my opinion, the review of related works in vague and hard to follow. I would have liked a more organised comparison.

- No intuition is given about why the \log d factor pops up in the analysis. This would help the reader assess to what extent this can be considered inherent to the completion setting or not.

Significance -------------- For the previous reasons, I find it hard to assess the significance.

Originality ------------- I believe that the results are new for the model and estimators considered, although they are very similar to other results.

The paper introduces a localised Gaussian width for the purposes of the analysis, and an inequality comparing the quantity to the Gaussian width based on chaining. I believe these results are novel, and could maybe be re-used for other purposes.

Quality --------- There are many typos:

line 113: "if one of the following conditions *is* satisfied"? line 167: "of all subsets *of* [d]"? line 333: "R(X) K \|X\|_*" example 4: do you mean the convex hull of the elliptope in the definition of R(\theta)?
Summary: The paper is interesting, but I do not find it clear enough as to why a separate analysis is required and how this compares to the very large body of estimators, models and bounds for estimation with general structured norms.

Submitted by Assigned_Reviewer_7

This paper studies the novel problem of matrix completion with a general norm regularization. Theoretical analyses are given which show that exact recover can be achieved if there are enough observed entries. Several potential applications are given and discussed.

Although several potential applications are presented in the paper (Examples 1 to 4), it seems to me that

it is still not clear how to interpret the results for those examples. For example in section 3.2, I see that the result is applied to the analysis of spectral k-support norm, but I cannot see the significance of such analysis.
Summary: This paper has interesting results on a general norm regularized matrix completion problem. This work extends the traditionally studied problem of low rank matrix completion and reveals promising research directions.

Author Feedback
Author rebuttal: We thank reviewers for detailed comments.

Assigned_Reviewer_2
Thanks for catching the errors. Will carefully fix the issues.
1. Error probability can be fixed to avoid the counter-intuitive 1/Omega term using concentration results from generic chaining (Mendelson et al., 07). Besides, as the sample complexity O(w_G^2logd)< w_G^4, technically samples over O(w_G^4) could be dropped at random.
2. Uniform sampling: A.1.1 can be fixed by bounding the number of collisions (prop3.3 in [29]) without significantly altering the analysis. Alternatively, we can redefine notation as ||P_O(X)||_F^2=\sum{ij\inO} Xij^2 rather than ||\sum_{ij\in O} Xij||_F^2.
3. We realize that outlier values in unobserved entries cannot be reconstructed in completion. Will elaborate on lowrank+P_O(sparse) matrices and their connection to spikiness constraint in the paper.
4a. The proof does follow. However, in our analysis, Th.2 is used on a subset of E_R with max norm restriction of O(1/d). Under this scaling, w(Y(E_R\capAco))<1 and the square term dominates.
4b. Th.2(and the bound by the reviewer) can be sharpened for many special cases. In fact, it can be shown that Lem9 is the tightest bound on w_(O,W). For subsets with l-infty restriction, Lem9 can directly be exploited for better bounds. Will modify Th.2 to make this connection explicit for l-infty restricted subsets.

Assigned_Reviewer_3
1. Proximal algorithms are attractive for norms with simple projections onto unit balls. For cases where R() is intractable, SDP relaxations from Sec.4.2 in [10] can possibly be extended to trade off sample complexity and computation.
2. References in Sec.3.1 have empirically demonstrated applications where structures beyond low rank are more effective. We develop general tools to theoretically analyze matrix completion under such structures. Detailed corollaries and simulated results for sample applications are beyond the scope of current version and will be included in a longer version.
3. For low rank, Th.1b recovers some standard results that have established lower bounds showing tightness. However, for general norms tightness of the bounds is not known.

Assigned_Reviewer_4
We will work on writing and clarify significance and connections to existing work.
1. Compared to analysis for Gaussian measurements, extension to matrix completion (MC) is non trivial as the RIP property of the measurement operator implicitly used in recovery analysis fail for MC (pl. refer [9] for discussion on this in the special case of low rank MC).
2. An important side result (Th.5) establishes a related RSC condition for localized samples--a result previously known only for nuclear norm.
3. We have a high level explanation for the log factor: signal recovery requires O(1) measurement of each independent parameter in the target. Under uniform subsampling (in finite dim.), there is a potential loading effect wherein, depending on the structure, there are more measurements of some independent parameters over the others. Log oversampling factor corrects for this effect. eg. in low rank MC with O(dr) degrees of freedom, log factor ensures O(r) samples per col whp. In analysis the log factor arises in showing concentration for RSC conditions.

Assigned_Reviewer_5
1. Will clarify 84-87. The results are comparable only beyond the noise levels where the first term in Th.1 dominate, and are weaker for lower noise levels.
2. Traditional incoherence can be thought of as a measure of spread of individual matrix entries along each degree of freedom in the target matrix (leading singular vectors). This notion is not immediately appropriate (or easily extendable) for arbitrary structures enforced by general norms. Thus, we generalize a simpler problem of approximate recovery.

Assigned_Reviewer_6
1. Success prob.: Omega< =d^2 always (d is dimension of matrix). Th.1 holds whp as long as wG^4 >Omega. Besides, this term can be fixed (Pl. refer response 1. for Assigned_Reviewer_2)
2. Th.1b: With tight bounds on R^*(noise), 1b potentially obtains consistent rates. For low rank, 1b matches rates of [27]. Ref. lines 330-336.
3. Th.1a: Constraint is based on non-matching l2 norm, which leads to weaker (but simpler) analysis [8,10,36].
4. Th.2: Lem9 gives a tight bound. Th.2 can be sharpened for special cases (Pl. refer response 4b. for Assigned_Reviewer_2)

Assigned_Reviewer_7
While unified analysis under norm regularization has been studied for sub-Gaussian measurements, extensions to matrix completion is non-trivial as the measurements involve localized sampling that violate RIP properties. Pl. refer response 1. to Assigned_Reviewer_4 for elaboration. Will work on the organization of the paper to enhance readability.

Assigned_Reviewer_8
Th.1 can be applied to understand sample complexity and error decay for robust matrix completion under general structures. Will include a conclusion in Sec.3.2 with implications of the theorem.